# A Theory of Training Parameter-Shared Quantum Neural Networks from a Bayesian Perspective

## Abstract

The objective function landscape of Quantum Neural Networks (QNNs) is both numerically and theoretically demonstrated to be highly non-convex, exhibiting numerous local optima. This raises an important question regarding the efficiency of training QNNs: can the optimization error systematically converge to a target threshold as the number of optimization iterations grows polynomially with the number of qubits $n$? In this work, we explore this question by proposing a theoretical framework from a Bayesian perspective. We focus on the trainability of Parameter-Shared QNNs (PS-QNNs), a widely used model for solving combinatorial optimization problems. Our first result shows that noise-free PS-QNNs with a depth of $\tilde{\mathcal{O}}\left(\sqrt{\log n}\right)$ can be trained efficiently. Furthermore, we demonstrate that if each quantum gate is influenced by a $q$-strength local Pauli channel, the noisy PS-QNN with a depth of $\mathcal{O}\left(\log n / \log(1/q)\right)$ can also be trained efficiently. These results provide valuable insights into the performance of QNNs, particularly in the context of the noisy intermediate-scale quantum era.

## 1 Introduction

Quantum Neural Networks (QNNs) (Cerezo et al., 2021a; Bharti et al., 2022; Huang et al., 2023), a hybrid quantum-classical machine learning model, have demonstrated success in various tasks on Noisy Intermediate-Scale Quantum (NISQ) devices, including classification (Farhi & Neven, 2018; Abohashima et al., 2020; Deng, 2021; Li & Deng, 2022; Song et al., 2024a; Wei et al., 2022; Li et al., 2022; Song et al., 2024b), regression (Kyriienko et al., 2021; Chen et al., 2022; Wu et al., 2023), generative learning (Zoufal et al., 2019; Situ et al., 2020; Nakaji & Yamamoto, 2021; Wu et al., 2024), and reinforcement learning (Chen et al., 2020; Lockwood & Si, 2020). A notable example of QNNs is the Parameter-Shared QNN (PS-QNN), which is also known as the quantum alternating operator ansatz (Farhi et al., 2014). This model is designed to solve combinatorial optimization problems that involve optimizing a quadratic function of binary variables (Lucas, 2014). As a hybrid quantum-classical model, the quantum component combines the quantum network and a problem-oriented Hamiltonian encoding the quadratic function to generate the PS-QNN objective function. Meanwhile, the classical component searches for the optimal solution of the PS-QNN objective function, employing either gradient-based classical optimization methods (Guerreschi & Smelyanskiy, 2017; Sweke et al., 2020; Stokes et al., 2020; Harrow & Napp, 2021; Koczor & Benjamin, 2022) or gradient-free classical optimization methods (Zhu et al., 2019; Self et al., 2021; Tibaldi et al., 2023) to iteratively update the parameters within the PS-QNN.

Understanding the trainability of the PS-QNN is crucial for assessing its potential quantum advantages in the NISQ era. The barren plateau phenomenon (McClean et al., 2018) during the training process is a significant topic, which directly relates to the sample complexity. In this context, the gradient function of the PS-QNN may vanish exponentially as the system size increases, indicating that obtaining an effective gradient function could demand an unmanageable level of sample complexity. Specifically, Ref. Cerezo et al. (2021b) proved that the gradient function induced by the $\mathcal{O}(\log n)$-depth PS-QNN and a local Hamiltonian is greater than $1/\text{poly}(n)$ when either the left or right slice of each block in the PS-QNN forms a local 1-design (Harrow & Low, 2009; Cerezo et al., 2021b). Meanwhile, similar findings also exist in the noisy PS-QNN scenario (Wang et al., 2021). In both cases, although the $\mathcal{O}(\log n)$-depth PS-QNN does not suffer from the barren plateau

Table 1: Comparisons between our works and related previous studies. Here, PL represents Polyak-Lojasiewicz, SGD represents Stochastic Gradient Descent, SMD represents Stochastic Mirror Descent, and BO represents Bayesian Optimization.

| Refs on Convergence Rate | Convexity | Environment | Optimization |
| --- | --- | --- | --- |
| Ref. Harrow & Napp (2021) | Strong Convexity | noise-free | SGD and SMD |
| Ref. Sweke et al. (2020) | PL inequality | noise-free | SGD |
| This work | No Convexity Assumptions | noise-free and noisy | BO |

| Refs on Sample Complexity | Assumptions | Environment | Trainability |
| --- | --- | --- | --- |
| Ref. McClean et al. (2018) | Global 2-design | noise-free | Exclude $\Omega(n)$-depth |
| Ref. Cerezo et al. (2021b) | Assumption 4.2 | noise-free | $\mathcal{O}(\log n)$ is unclear |
| Ref. Wang et al. (2021) | Assumption 5.3 | noisy | Exclude $\Omega(\log n)$-depth |
| This work | Assumptions 4.2, 5.3 | noise-free and noisy | $\tilde{\mathcal{O}}(\log n)$ is trainable |

phenomenon, the manageable sample complexity does not necessarily imply the effective training of the corresponding PS-QNN. Therefore, further theoretical analysis of the convergence performance of the classical optimization method employed in the PS-QNN is necessary.

Previous studies assumed either strong convexity (Harrow & Napp, 2021) or the satisfaction of the Polyak-Lojasiewicz inequality (Sweke et al., 2020). In these contexts, the convergence performance of stochastic gradient descent, a gradient-based optimization method, has been theoretically analyzed. However, in realistic scenarios, the PS-QNN objective function landscape is generally non-convex with numerous local optima (Shaydulin et al., 2019; Huembeli & Dauphin, 2021).

In this work, considering the PS-QNN objective function and the sample drawn from the Gaussian process with the Matern covariance function share similar high-order differentiability properties (Williams & Rasmussen, 2006; Kanagawa et al., 2018; Bouland et al., 2019; Wu et al., 2025; Fontana et al., 2025), we naturally regard the PS-QNN objective function as a sample drawn from this Gaussian process, referred to as the Bayesian perspective. This perspective offers valuable insights into the PS-QNN objective function, which may be more realistic than the assumptions used in previous studies. Subsequently, under this perspective, we theoretically analyze the convergence performance of Bayesian Optimization (BO) (Snoek et al., 2012; Shahriari et al., 2015; Frazier, 2018), a gradient-free global optimization method, in both the noise-free PS-QNN and the noisy PS-QNN exposed to local Pauli channels (Wang et al., 2021; Quek et al., 2024). Specifically, we start by analyzing the continuity property of the PS-QNN objective function. By leveraging this property, we establish a theoretical limit on the network depth, guaranteeing the convergence of the optimization error to a target threshold as the number of optimization iterations scales polynomially with the system size. Finally, we provide the efficient trainable network depth that satisfies the above constraints and avoids the barren plateau phenomenon, such that the PS-QNN has both acceptable sample complexity and optimization iteration complexity.

Our theoretical analysis shows that when either the left or right slice of each block in the $n$-qubit noise-free PS-QNN forms a local 1-design, the network with a depth of $\tilde{\mathcal{O}}(\sqrt{\log n})$ can be trained efficiently. Furthermore, when each quantum gate is affected by a local Pauli channel with the strength $q$ ranging from $1/\text{poly}(n)$ to $0.1$, we demonstrate that the noisy PS-QNN with a depth of $\mathcal{O}(\log n / \log(1/q))$ can also be trained efficiently for the Maximum Cut problem on an unweighted regular graph. These findings provide important theoretical insights for exploring potential quantum advantages, particularly within the NISQ era. For easy reference, connections between our work and relevant prior studies are summarized in Table 1.

## 2 PS-QNNs FOR SOLVING COMBINATORIAL OPTIMIZATION PROBLEMS

In theoretical computational science, combinatorial optimization problems encompass a wide range of typical problems, such as Maximum Cut, Maximum Independent Set, and Graph Coloring (Gross

et al., 2018). These problems define their constraints as clauses, with a candidate solution represented by a specific assignment of the corresponding binary variables. The objective of these problems is to find an optimal assignment that maximizes the number of satisfied clauses. In other words, solving a combinatorial optimization problem can be reformulated as optimizing a quadratic function involving binary variables. However, finding the exact solution is widely recognized as an NP-hard problem (Garey & Johnson, 2002). Consequently, an alternative approach is to seek an approximate solution. Inspired by the quantum annealing process (Kadowaki & Nishimori, 1998), the PS-QNN was proposed and applied to solve combinatorial optimization problems. Although the prospects of achieving quantum advantages through the PS-QNN remain unclear, it provides a simple paradigm for optimization that can be implemented on near-term quantum devices.

Given a specific combinatorial optimization problem with $n$ binary variables and $\mathcal{C}$ clauses, the PS-QNN starts by constructing a problem-oriented Hamiltonian $H_1$, a mixing Hamiltonian $H_2$, and $2p$ variational parameters $\boldsymbol{\theta} = [\theta_{1,1}, \theta_{1,2}, \cdots, \theta_{p,1}, \theta_{p,2}]^{\mathsf{T}}$. Specifically, the problem-oriented Hamiltonian $H_1$ is a linear combination of $\mathcal{C}$ Pauli strings

$$H_1 = \sum_{c=1}^{\mathcal{C}} \gamma_c P_c^n, \tag{1}$$

where $\gamma_c \in \mathbb{R}$ and $P_c^n \in \{\mathbb{I}, \sigma^z\}^{\otimes n}$ with $\sigma^z$ is the Pauli $Z$ operator. The typical form of the mixing Hamiltonian $H_2$ is the transverse field Hamiltonian

$$H_2 = \sum_{i=1}^{n} \sigma_i^x, \tag{2}$$

where $\sigma_i^x$ is the Pauli $X$ operator acting on the $i$-th qubit. Subsequently, by iteratively applying $H_1$ and $H_2$ to the initial state $\rho$ for $p$ rounds, the noise-free PS-QNN objective function $f(\boldsymbol{\theta})$ in the absence of quantum gate noise is given by the following expectation value

$$f(\boldsymbol{\theta}) = \text{Tr}\left[H_1 U(\boldsymbol{\theta}) \rho U^{\dagger}(\boldsymbol{\theta})\right], \tag{3}$$

where $\rho = (|+\rangle\langle+|)^{\otimes n}$ denotes the uniform superposition over computational basis states and the noise-free PS-QNN

$$U(\boldsymbol{\theta}) = \prod_{j=1}^{p} \prod_{l=1}^{2} U_{j,l}(\theta_{j,l}) \tag{4}$$

with $U_{j,l}(\theta_{j,l}) = \exp(-i\theta_{j,l} H_l)$ for $(j,l) \in [p] \times [2]$. The statistical estimation of $f(\boldsymbol{\theta})$ can be achieved by repeating the aforementioned process with identical parameters and computational basis measurements. After defining $f(\boldsymbol{\theta})$, the next step involves iteratively updating $\boldsymbol{\theta}$ within $U(\boldsymbol{\theta})$ through classical optimization methods to maximize $f(\boldsymbol{\theta})$ and obtain the global maximum point

$$\boldsymbol{\theta}^* = \arg\max_{\boldsymbol{\theta} \in \mathcal{D}} f(\boldsymbol{\theta}), \tag{5}$$

where the domain $\mathcal{D} = [0, 2\pi]^{2p}$.

The classical optimization method plays a crucial role in this process. Finding $\boldsymbol{\theta}^*$ may be intractable due to the non-convex landscape of $f(\boldsymbol{\theta})$ and the presence of numerous local optima. Thus, it is pivotal to determine the appropriate classical optimization method that can efficiently find a better approximation of $\boldsymbol{\theta}^*$. In the following section, we illustrate the utilization of BO to accomplish this optimization task and provide a theoretical analysis of the trainability of PS-QNNs.

## 3 OPTIMIZING PS-QNNS THROUGH BO

BO is designed for gradient-free global optimization. It is particularly suitable in situations where estimating the objective function is computationally expensive and the convexity property of the objective function is uncertain (Snoek et al., 2012; Shahriari et al., 2015; Frazier, 2018). BO comprises two essential components: (i) a statistical model, usually the Gaussian process (Williams & Rasmussen, 2006), that generates a posterior distribution conditioned on a prior distribution and a collection of observations of the objective function. (ii) an acquisition function that utilizes the current posterior distribution for the objective function to determine the location of the next point. In the context of PS-QNNs, we present a comprehensive introduction to BO, focusing on the Gaussian process and the acquisition function.

## 3.1 GAUSSIAN PROCESS

A Gaussian process is a collection of random variables, where any subset forms a multivariate Gaussian distribution. In the optimization task described by Eq. 5, the random variables correspond to the values of the noise-free PS-QNN objective function $f(\boldsymbol{\theta})$ at points $\boldsymbol{\theta} \in \mathcal{D}$. A Gaussian process, serving as a distribution for $f(\boldsymbol{\theta})$, is fully determined by its mean function $\mu(\boldsymbol{\theta})$ and covariance function $k(\boldsymbol{\theta}, \boldsymbol{\theta}')$. Specifically, $\mu(\boldsymbol{\theta})$ specifies the mean value of $f(\boldsymbol{\theta})$ at any point $\boldsymbol{\theta}$, while $k(\boldsymbol{\theta}, \boldsymbol{\theta}')$ determines the covariance between $f(\boldsymbol{\theta})$ and $f(\boldsymbol{\theta}')$ at any two points $\boldsymbol{\theta}$ and $\boldsymbol{\theta}'$. The Gaussian process is denoted as

$$f(\boldsymbol{\theta}) \sim \mathcal{GP}(\mu(\boldsymbol{\theta}), k(\boldsymbol{\theta}, \boldsymbol{\theta}')), \tag{6}$$

where $\mu(\boldsymbol{\theta}) = \mathbb{E}[f(\boldsymbol{\theta})]$ and $k(\boldsymbol{\theta}, \boldsymbol{\theta}') = \mathbb{E}[(f(\boldsymbol{\theta}) - \mu(\boldsymbol{\theta}))(f(\boldsymbol{\theta}') - \mu(\boldsymbol{\theta}'))]$. It is commonly assumed that the prior mean function $\mu(\boldsymbol{\theta}) = 0$. Additionally, the prior covariance function $k(\boldsymbol{\theta}, \boldsymbol{\theta}')$ is commonly chosen from some notable covariance functions, such as the Matern covariance function $k_{\text{Matern}-\nu}(\boldsymbol{\theta}, \boldsymbol{\theta}')$, whose specific form is provided in Appendix A.1.

Suppose we have the following accumulated observations after $t - 1$ steps of BO

$$\mathcal{S}_{t-1} = \{(\boldsymbol{\theta}_1, y(\boldsymbol{\theta}_1)), \cdots, (\boldsymbol{\theta}_{t-1}, y(\boldsymbol{\theta}_{t-1}))\}, \tag{7}$$

where $y(\boldsymbol{\theta}_i)$ denotes the estimation of $f(\boldsymbol{\theta}_i)$ for $i \in [t-1]$. In each step $i$, the measurements are taken to ensure that $y(\boldsymbol{\theta}_i) = f(\boldsymbol{\theta}_i) + \xi_i^{\text{noise}}$, where $\xi_i^{\text{noise}} \sim N(0, 1/4M)$ is independent and identically distributed Gaussian noise with $M$ representing the fixed number of measurements [1]. Conditioned on $\mathcal{S}_{t-1}$, the distribution for $f(\boldsymbol{\theta})$ is a Gaussian process with the posterior mean function $\mu_{t-1}(\boldsymbol{\theta}) = \mathbb{E}[f(\boldsymbol{\theta})|\mathcal{S}_{t-1}]$ and the posterior covariance function $k_{t-1}(\boldsymbol{\theta}, \boldsymbol{\theta}') = \mathbb{E}[(f(\boldsymbol{\theta}) - \mu(\boldsymbol{\theta}))(f(\boldsymbol{\theta}') - \mu(\boldsymbol{\theta}'))|\mathcal{S}_{t-1}]$. These are specified as follows

$$\begin{aligned} \mu_{t-1}(\boldsymbol{\theta}) &= \boldsymbol{k}_{t-1}(\boldsymbol{\theta})^{\mathsf{T}}(\boldsymbol{K}_{t-1} + \boldsymbol{I}_{t-1}/4M)^{-1}\boldsymbol{y}_{t-1}, \\ k_{t-1}(\boldsymbol{\theta}, \boldsymbol{\theta}') &= k_{\text{Matern}-\nu}(\boldsymbol{\theta}, \boldsymbol{\theta}') - (\boldsymbol{k}_{t-1}(\boldsymbol{\theta})^{\mathsf{T}}(\boldsymbol{K}_{t-1} + \boldsymbol{I}_{t-1}/4M)^{-1}\boldsymbol{k}_{t-1}(\boldsymbol{\theta}')), \end{aligned} \tag{8}$$

where $\boldsymbol{k}_{t-1}(\boldsymbol{\theta}) = [k_{\text{Matern}-\nu}(\boldsymbol{\theta}, \boldsymbol{\theta}_1) \cdots k_{\text{Matern}-\nu}(\boldsymbol{\theta}, \boldsymbol{\theta}_{t-1})]^{\mathsf{T}}$, the positive definite covariance matrix $\boldsymbol{K}_{t-1} = [k_{\text{Matern}-\nu}(\boldsymbol{\theta}_i, \boldsymbol{\theta}_j)]_{\boldsymbol{\theta}_i, \boldsymbol{\theta}_j \in \mathcal{A}_{t-1}}$ with the accumulated points $\mathcal{A}_{t-1} = \{\boldsymbol{\theta}_1, \cdots, \boldsymbol{\theta}_{t-1}\}$ from the previous $t - 1$ steps, and $\boldsymbol{y}_{t-1} = [y(\boldsymbol{\theta}_1) \cdots y(\boldsymbol{\theta}_{t-1})]^{\mathsf{T}}$. The posterior variance of $f(\boldsymbol{\theta})$ is denoted as $\sigma_{t-1}^2(\boldsymbol{\theta}) = k_{t-1}(\boldsymbol{\theta}, \boldsymbol{\theta})$.

## 3.2 ACQUISITION FUNCTION

In the $t$-th step of BO, the acquisition function utilizes $\mathcal{S}_{t-1}$ to guide the search towards the next point $\boldsymbol{\theta}_t$, aiming to converge to the global maximum point $\boldsymbol{\theta}^*$ of $f(\boldsymbol{\theta})$. This procedure is accomplished by maximizing the acquisition function over the domain $\mathcal{D}$. Specifically, the design of the acquisition function should consider both exploration (sampling in regions of high uncertainty) and exploitation (sampling in regions likely to yield high function values). The upper confidence bound $\text{UCB}_t(\boldsymbol{\theta})$, which is a commonly employed acquisition function, is defined as

$$\text{UCB}_t(\boldsymbol{\theta}) = \mu_{t-1}(\boldsymbol{\theta}) + \sqrt{\eta_t}\sigma_{t-1}(\boldsymbol{\theta}), \tag{9}$$

and the next point $\boldsymbol{\theta}_t$ is selected as

$$\boldsymbol{\theta}_t = \arg\max_{\boldsymbol{\theta} \in \mathcal{D}} \text{UCB}_t(\boldsymbol{\theta}), \tag{10}$$

where $\mu_{t-1}(\boldsymbol{\theta})$ and $\sigma_{t-1}(\boldsymbol{\theta})$ denote the posterior mean function and the posterior standard deviation respectively, as defined in Eq. 8, and $\eta_t \geq 0$ represents a time-dependent scaling parameter. Subsequently, the accumulated observations are updated as $\mathcal{S}_t = \{(\boldsymbol{\theta}_1, y(\boldsymbol{\theta}_1)), \cdots, (\boldsymbol{\theta}_t, y(\boldsymbol{\theta}_t))\}$, and the posterior distribution for $f(\boldsymbol{\theta})$ is updated based on $\mathcal{S}_t$.

---

[1] Let $M$ be the fixed number of measurements and $y_j(\boldsymbol{\theta})$ be the one-shot measurement result for $j \in [M]$. According to Central Limit Theorem (Fischer, 2011), for a sufficiently large $M$, we have $\frac{1}{M}\sum_{j=1}^{M} y_j(\boldsymbol{\theta}) = \mu + \frac{\sigma}{\sqrt{M}}Y$, where $Y \sim N(0, 1)$. Here, $\mu$ and $\sigma$ represent the mean value and standard deviation of $y_j(\boldsymbol{\theta})$ for $j \in [M]$. Consequently, we find $\xi^{\text{noise}} = \frac{1}{M}\sum_{j=1}^{M} y_j(\boldsymbol{\theta}) - \mu = \frac{\sigma}{\sqrt{M}}Y$. Thus, the result $\xi^{\text{noise}} \sim N(0, \sigma^2/M)$ holds. Considering $\sigma^2 \leq 1/4$, it is reasonable to assume that $\xi^{\text{noise}} \sim N(0, 1/4M)$.

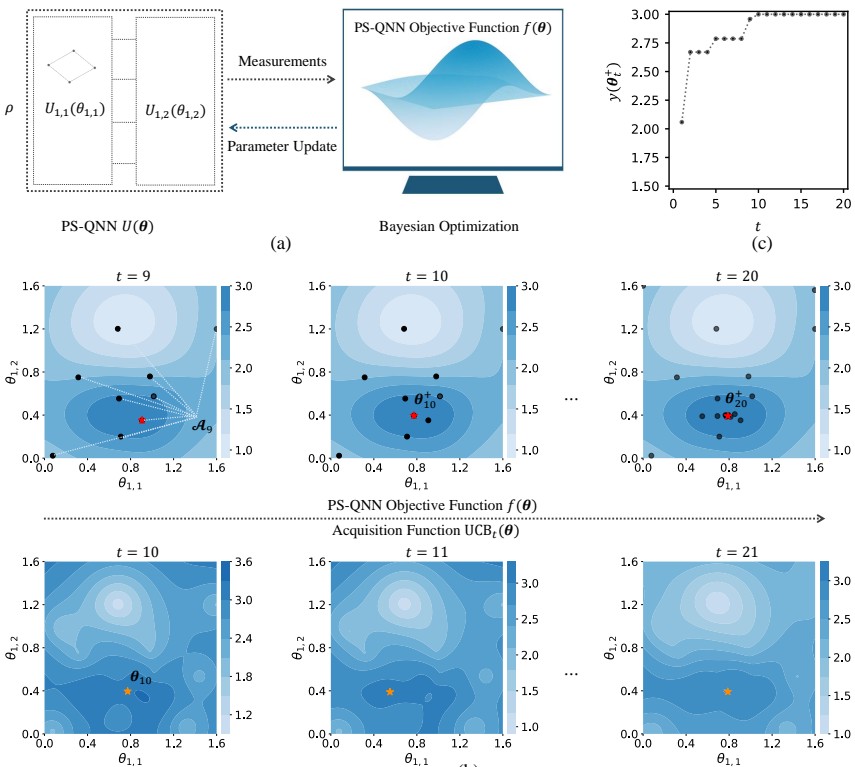

Figure 1: Numerical performance of PS-QNNs for solving Maximum Cut problems through BO. (a) Structure of PS-QNNs on a Maximum Cut graph instance. For a 2-regular graph with 4 vertices, the quantum subroutine prepares parameterized quantum states using a 1-layer noise-free PS-QNN $U(\boldsymbol{\theta})$. It estimates the corresponding two-dimensional noise-free PS-QNN objective function $f(\boldsymbol{\theta})$ with $\boldsymbol{\theta} = (\theta_{1,1}, \theta_{1,2})$ through multi-round measurements. BO iteratively updates $\boldsymbol{\theta}$ within $U(\boldsymbol{\theta})$ until reaching the predetermined number of steps, ultimately providing the approximation of the maximum point. (b) Detailed optimization steps of BO on the given Maximum Cut graph instance. In the 10th step, the acquisition function $\mathrm{UCB}_{10}(\boldsymbol{\theta})$ are calculated based on the accumulated points $\mathcal{A}_9$ from the previous 9 steps. The next point $\boldsymbol{\theta}_{10}$ is selected by maximizing $\mathrm{UCB}_{10}(\boldsymbol{\theta})$, and the current approximation of the maximum point $\boldsymbol{\theta}_{10}^+$ is updated to the best point selected in the previous 10 steps. Finally, after 20 steps, the final approximation of the maximum point $\boldsymbol{\theta}_{20}^+$ is returned. (c) The estimation of the function value at the approximation of the maximum point $y(\boldsymbol{\theta}_t^+)$ as a function of the step $t$.

The aforementioned process is repeated for a predetermined number of steps $T$. The best point selected in the previous $T$ steps represents the approximation of the maximum point $\boldsymbol{\theta}_T^+$. Specifically,

$$\boldsymbol{\theta}_T^+ = \arg \max_{\boldsymbol{\theta} \in \mathcal{A}_T} y(\boldsymbol{\theta}), \tag{11}$$

where $\mathcal{A}_T = \{\boldsymbol{\theta}_1, \cdots, \boldsymbol{\theta}_T\}$ represents the accumulated points from the previous $T$ steps. Figure 1 illustrates the numerical performance of PS-QNNs for solving the Maximum Cut problem on a specific graph instance using BO. The numerical results regarding the performance of BO on diverse graph structures and the comparison between BO and Gradient Descent(GD) can be found in Appendices F.1 and F.2, respectively. In conclusion, the Gaussian process is widely preferred as the statistical model in BO due to its flexibility and capacity to model complex functions. It offers a powerful framework for modeling the objective function by capturing both the mean and uncertainty associated with observations of the objective function. This enables BO to make informed decisions regarding the location of the next point. Subsequently, the upper confidence bound balances the trade-off between exploration and exploitation and selects the next point based on the current knowledge provided by the Gaussian process. In summary, the combination of the Gaussian process as the statistical model and the upper confidence bound as the acquisition function consti-

tutes the core of the BO framework, enabling efficient global optimization in the absence of gradient information.

## 4 MAIN RESULTS I: ANALYZING THE TRAINABILITY OF THE NOISE-FREE PS-QNN

Our main focus is to theoretically investigate the trainability of the $n$-qubit noise-free PS-QNN $U(\boldsymbol{\theta})$ using BO. The optimization error $r_T$ after $T$ steps of executing BO is defined as the difference in function values between the global maximum point $\boldsymbol{\theta}^*$ and the approximation of the maximum point $\boldsymbol{\theta}_T^+$ in the previous $T$ steps. It is given by

$$r_T = f(\boldsymbol{\theta}^*) - f(\boldsymbol{\theta}_T^+), \tag{12}$$

where $\boldsymbol{\theta}_T^+ = \arg\max_{\boldsymbol{\theta} \in \mathcal{A}_T} f(\boldsymbol{\theta})$ with the accumulated points $\mathcal{A}_T = \{\boldsymbol{\theta}_1, \cdots, \boldsymbol{\theta}_T\}^2$.

**Definition 4.1** (Effective network depth and parameter dimension). Given a $p$-depth PS-QNN $U(\boldsymbol{\theta})$ (with $p$ repeated variational blocks), we define the maximum $p$ enabling $r_T \leq \epsilon$ after $T = \text{poly}(n)$ steps as the effective network depth of $U(\boldsymbol{\theta})$. In the context of PS-QNNs, the effective parameter dimension and the network depth are equivalent.

Hence, we can directly explore the *effective parameter dimension* $p$ of $U(\boldsymbol{\theta})$ in subsequent analysis. In this work, we adopt the following widely accepted assumption.

**Assumption 4.2** (Harrow & Low (2009); Cerezo et al. (2021b)). Given an $n$-qubit noise-free PS-QNN

$$U(\boldsymbol{\theta}) = \prod_{j=1}^{p} \prod_{l=1}^{2} U_{j,l}(\theta_{j,l}), \tag{13}$$

each block $U_{j,l}(\theta_{j,l}) = U_+^{(j,l)} U_-^{(j,l)}$ for $(j,l) \in [p] \times [2]$, where $U_-^{(j,l)}$ is independent to $U_+^{(j,l)}$, and at least one of them forms a local 1-design.

The scenario mentioned above for investigating the trainability of $U(\boldsymbol{\theta})$ using BO is described in detail in Figure 2(a). Assuming that Assumption 4.2 holds, we first explore the Lipschitz continuity of the corresponding noise-free PS-QNN objective function $f(\boldsymbol{\theta})$. Additionally, we establish a theoretical limit on $p$ that ensures achieving $r_T \leq \epsilon$ within $T = \text{poly}(n)$ steps. The following sections provide a comprehensive introduction.

### 4.1 CONTINUITY PROPERTY OF THE NOISE-FREE PS-QNN OBJECTIVE FUNCTION

Now, we will show that Assumption 4.2 results in a quantum analog of the Lipschitz continuity property about the noise-free PS-QNN objective function $f(\boldsymbol{\theta})$.

**Lemma 4.3.** *Assuming that Assumption 4.2 holds, let $f(\boldsymbol{\theta}) : \mathcal{D} = [0, 2\pi]^{2p} \mapsto \mathbb{R}$ be the noise-free PS-QNN objective function (Eq. 3). Given a failure probability $\delta \in (0, 1)$, for any $\boldsymbol{\theta}, \boldsymbol{\theta}' \in \mathcal{D}$, the relationship*

$$|f(\boldsymbol{\theta}) - f(\boldsymbol{\theta}')| \leq \sqrt{\mathbb{V}_{\boldsymbol{\theta}}[\partial_a f(\boldsymbol{\theta})]/\delta} \|\boldsymbol{\theta} - \boldsymbol{\theta}'\|_1 \tag{14}$$

*is valid with a success probability of at least $1 - \delta$, where $\mathbb{V}_{\boldsymbol{\theta}}[\partial_a f(\boldsymbol{\theta})]$ is the variance of the partial derivative $\partial_a f(\boldsymbol{\theta})$ with index $a = \arg\max_{j \in [2p]}(\sup_{\boldsymbol{\theta} \in \mathcal{D}} |\partial_j f(\boldsymbol{\theta})|)$.*

Proof details of Lemma 4.3 can be found in Appendix B. In addition to Assumption 4.2, we provide the following remark based on the differentiable property of $f(\boldsymbol{\theta})$.

*Remark* 4.4. Given the differentiable property of the noise-free PS-QNN objective function $f(\boldsymbol{\theta})$ (Bouland et al., 2019; Wu et al., 2025), we can consider it as a sample drawn from a Gaussian process with the Matern prior covariance function $k_{\text{Matern}-\nu}(\boldsymbol{\theta}, \boldsymbol{\theta}')$ (Appendix A.1), as this Gaussian process allows us to model high-order differentiable functions (Williams & Rasmussen, 2006; Kanagawa et al., 2018).

---

[2]We assume by default that the estimations $\{y(\boldsymbol{\theta})\}_{\boldsymbol{\theta} \in \mathcal{A}_T}$ from the previous $T$ steps contain sufficient information about the noise-free PS-QNN objective function values $\{f(\boldsymbol{\theta})\}_{\boldsymbol{\theta} \in \mathcal{A}_T}$ with the accumulated points $\mathcal{A}_T = \{\boldsymbol{\theta}_1, \ldots, \boldsymbol{\theta}_T\}$. Therefore, it is reasonable to define the approximation of the maximum point as $\boldsymbol{\theta}_T^+ = \arg\max_{\boldsymbol{\theta} \in \mathcal{A}_T} f(\boldsymbol{\theta})$, even though it was originally defined as $\boldsymbol{\theta}_T^+ = \arg\max_{\boldsymbol{\theta} \in \mathcal{A}_T} y(\boldsymbol{\theta})$.

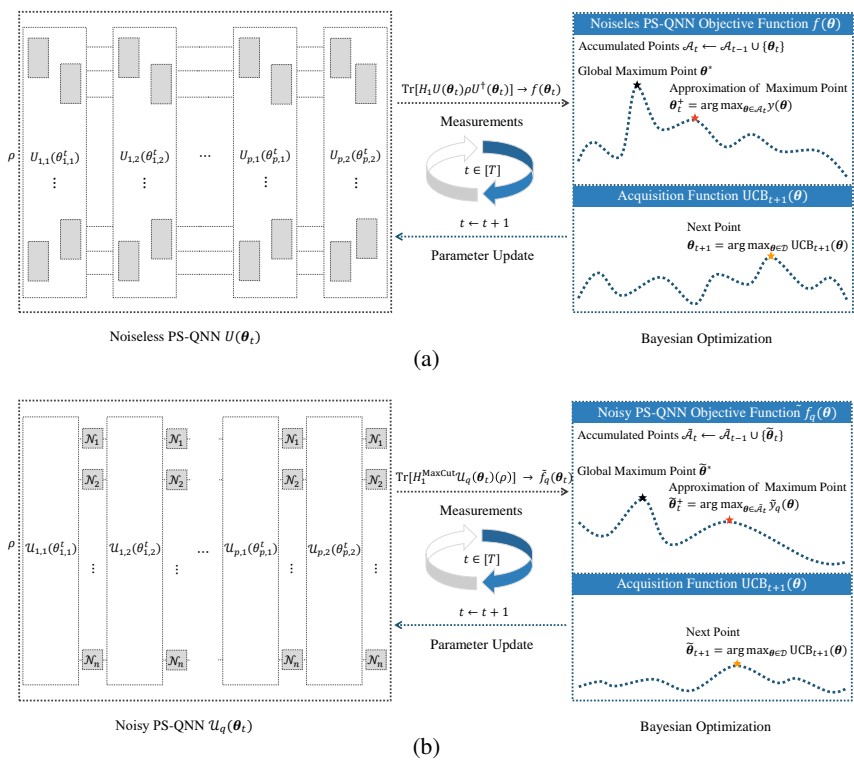

Figure 2: Scenarios to investigate the trainability of PS-QNNs. (a) Optimization of the $n$-qubit noise-free PS-QNN using BO. In the $t$-th step of the quantum subroutine, the noise-free PS-QNN $U(\boldsymbol{\theta}_t)$ prepares parameterized quantum states, where either the left or right slice of each block in $U(\boldsymbol{\theta}_t)$ forms a local 1-design. Next, the noise-free PS-QNN objective function $f(\boldsymbol{\theta}_t)$ is estimated through the fixed number of measurements $M$, yielding the estimation $y(\boldsymbol{\theta}_t)$. In the classical subroutine, BO utilizes the acquisition function $\mathrm{UCB}_{t+1}(\boldsymbol{\theta})$ to select the next point $\boldsymbol{\theta}_{t+1}$ based on the current knowledge provided by the Gaussian process. Afterwards, the variational parameters are updated. This process is repeated for a predetermined number of steps $T$, and the best point selected in the previous $T$ steps represents the approximation of the maximum point $\boldsymbol{\theta}_T^+$. (b) Optimization of the $n$-qubit noisy PS-QNN using BO. In the $t$-th step of the quantum subroutine, the noisy PS-QNN $\mathcal{U}_q(\boldsymbol{\theta}_t)$ prepares parameterized quantum states, where a noise channel $\mathcal{N} = \bigotimes_{i=1}^{n} \mathcal{N}_i$ exists between any two blocks in the network, and $\mathcal{N}_i$ represents a local Pauli channel acting on the $i$-th qubit. Next, the noisy PS-QNN objective function $\tilde{f}_q(\boldsymbol{\theta}_t)$ is estimated through the fixed number of measurements $M$, yielding the estimation $\tilde{y}_q(\boldsymbol{\theta}_t)$. In the classical subroutine, BO iteratively updates the variational parameters using the same optimization process for $T$ steps and eventually returns the approximation of the maximum point $\tilde{\boldsymbol{\theta}}_T^+$. The presence of noise may result in a flatter landscape with fewer local optima for $\tilde{f}_q(\boldsymbol{\theta})$ compared to $f(\boldsymbol{\theta})$.

## 4.2 Effective Parameter Dimension of the Noise-free PS-QNN

Assuming that Assumption 4.2 holds and using the result of Lemma 4.3, we establish a theoretical limit on the *effective parameter dimension* $p$ (Definition 4.1) through the perspective of the Bayesian approach.

**Theorem 4.5** (Informal)**.** *Given a constant threshold $\epsilon$ and an $n$-qubit noise-free PS-QNN $U(\boldsymbol{\theta})$ (Eq. 13) that satisfies Assumption 4.2, run BO for $T = \mathrm{poly}(n^{1/\epsilon^2})$ steps, where a predefined scaling parameter $\eta_t$ for the acquisition function $\mathrm{UCB}_t(\boldsymbol{\theta})$ (Eq. 9) is used in each step $t$. If the parameter dimension*

$$p \le \tilde{\mathcal{O}}\left(\sqrt{\log n}\right), \tag{15}$$

*then the optimization error $r_T$ (Eq. 12) satisfies $r_T \le \epsilon$ with high success probability.*

The formal statement and corresponding proof details of Theorem 4.5 are provided in Appendix C. The numerical validation of this theoretical result is provided in Appendix F.3. In summary, we consider the $n$-qubit noise-free PS-QNN objective function $f(\boldsymbol{\theta})$ as a sample drawn from a Gaussian process with the Matern covariance function $k_{\mathrm{Matern}-\nu}(\boldsymbol{\theta}, \boldsymbol{\theta}')$, leveraging its high-order differentiable property. We then investigate the trainability of the corresponding noise-free PS-QNN $U(\boldsymbol{\theta})$ through this perspective. Based on Assumption 4.2 that either the left or right slice of each block in $U(\boldsymbol{\theta})$ forms a local 1-design, we demonstrate that $U(\boldsymbol{\theta})$ with a parameter dimension $p$ of $\tilde{\mathcal{O}}(\sqrt{\log n})$ can be trained efficiently using BO.

## 5 MAIN RESULTS II: ANALYZING THE TRAINABILITY OF THE NOISY PS-QNN

After exploring the trainability of the noise-free PS-QNN $U(\boldsymbol{\theta})$ through BO, we will proceed to investigate its theoretical performance in a practical scenario. This scenario involves the Maximum Cut problem on an unweighted regular graph, where $U(\boldsymbol{\theta})$ is affected by local Pauli channels. For the sake of clarity, we begin by presenting the definitions of the Maximum Cut problem and the local Pauli channel.

**Definition 5.1** (Maximum Cut problem). Considering an unweighted $d$-regular graph $G = (V, E)$ with the vertices set $V = \{v_1, \cdots, v_n\}$ and the edges set $E = \{e_{i,j}\}$, the Maximum Cut problem aims at dividing all vertices into two disjoint sets such that maximizing the number of edges that connect the two sets. In the context of PS-QNNs, the problem-oriented Hamiltonian $H_1^{\mathrm{MaxCut}}$ is defined as

$$H_1^{\mathrm{MaxCut}} = \frac{1}{2} \sum_{e_{i,j} \in E} (\mathbb{I}^{\otimes n} - \sigma_i^z \sigma_j^z). \tag{16}$$

**Definition 5.2** (Local Pauli channel). Let $\mathcal{N}_i$ denote a local Pauli channel acting on each qubit $i$. The action of $\mathcal{N}_i$ on a local Pauli operator $\sigma \in \{\sigma^x, \sigma^y, \sigma^z\}$ can be expressed as

$$\mathcal{N}_i(\sigma) = q_\sigma \sigma, \tag{17}$$

where $q_{\sigma^x}, q_{\sigma^y}, q_{\sigma^z} \in (-1, 1)$. The noise strength in this model is represented by a single parameter $q = \sqrt{\max_{\sigma \in \{\sigma^x, \sigma^y, \sigma^z\}} |q_\sigma|}$.

Due to imperfections in quantum devices, we assume that each quantum gate is affected by a $q$-strength local Pauli channel, and the effects of these noises are postponed until the end of each block in $U(\boldsymbol{\theta})$. This assumption is reasonable, as it has been employed in Ref. Wang et al. (2021) and demonstrated to hold true in Clifford circuits (Quek et al., 2024). Below is a detailed and precise description of this assumption.

**Assumption 5.3.** Given the $q$-strength local Pauli channel $\mathcal{N}_i$ (Eq. 17) which is gate-independent and time-invariant, the $n$-qubit noisy PS-QNN is given by

$$\mathcal{U}_q(\boldsymbol{\theta}) = \bigcirc_{j=1}^{p} \bigcirc_{l=1}^{2} \left( \mathcal{N} \circ \mathcal{U}_{j,l}(\theta_{j,l}) \right), \tag{18}$$

where $\mathcal{N} = \bigotimes_{i=1}^{n} \mathcal{N}_i$ is the noise channel and $\mathcal{U}_{j,l}(\theta_{j,l})$ is the channel that implements the unitary $U_{j,l}(\theta_{j,l})$ for $(j, l) \in [p] \times [2]$.

Assuming that Assumption 5.3 holds, the $n$-qubit noisy PS-QNN objective function $\tilde{f}_q(\boldsymbol{\theta})$ with $q$-strength local Pauli channels is given by

$$\tilde{f}_q(\boldsymbol{\theta}) = \mathrm{Tr} \left[ H_1^{\mathrm{MaxCut}} \mathcal{U}_q(\boldsymbol{\theta})(\rho) \right], \tag{19}$$

where $H_1^{\mathrm{MaxCut}}$ is the problem-oriented Hamiltonian about the Maximum Cut problem (Eq. 16), $\mathcal{U}_q(\boldsymbol{\theta})$ is the noisy PS-QNN (Eq. 18) and $\rho = (|+\rangle\langle+|)^{\otimes n}$ is the initial state. Now, the optimization error $\tilde{r}_T$ after $T$ steps of executing BO is defined as the difference in function values between the global maximum point $\tilde{\boldsymbol{\theta}}^*$ and the approximation of the maximum point $\tilde{\boldsymbol{\theta}}_T^+$ in the previous $T$ steps. Specifically,

$$\tilde{r}_T = \tilde{f}_q(\tilde{\boldsymbol{\theta}}^*) - \tilde{f}_q(\tilde{\boldsymbol{\theta}}_T^+), \tag{20}$$

where $\tilde{\boldsymbol{\theta}}_T^+ = \arg\max_{\boldsymbol{\theta} \in \tilde{\mathcal{A}}_T} \tilde{f}_q(\boldsymbol{\theta})$ with the accumulated points $\tilde{\mathcal{A}}_T = \{\tilde{\boldsymbol{\theta}}_1, \cdots, \tilde{\boldsymbol{\theta}}_T\}$. Figure 2(b) provides a detailed description of the scenario mentioned above for investigating the trainability of $\mathcal{U}_q(\boldsymbol{\theta})$ using BO. In the following sections, we first explore the Lipschitz continuity of $\tilde{f}_q(\boldsymbol{\theta})$.

## 5.1 CONTINUITY PROPERTY OF THE NOISY PS-QNN OBJECTIVE FUNCTION

Now, we will show that Assumption 5.3 results in a quantum analog of the Lipschitz continuity property about the noisy PS-QNN objective function $\tilde{f}_q(\boldsymbol{\theta})$.

**Lemma 5.4.** *Assuming that Assumption 5.3 holds and considering the Maximum Cut problem on an unweighted $d$-regular graph with $n$ vertices, let $\tilde{f}_q(\boldsymbol{\theta}) : \mathcal{D} = [0, 2\pi]^{2p} \mapsto \mathbb{R}$ be the noisy PS-QNN objective function with $q$-strength local Pauli channels (Eq. 19). For any $\boldsymbol{\theta}, \boldsymbol{\theta}' \in \mathcal{D}$, the relationship*

$$\left| \tilde{f}_q(\boldsymbol{\theta}) - \tilde{f}_q(\boldsymbol{\theta}') \right| \leq d^3 n^{7/2} q^{(d+1)p} \|\boldsymbol{\theta} - \boldsymbol{\theta}'\|_1 \tag{21}$$

*is valid, where the strength $q \in (0, 1)$.*

Proof details of Lemma 5.4 can be found in Appendix D. In addition to Assumption 5.3, we provide the following remark based on the differentiable property of $\tilde{f}_q(\boldsymbol{\theta})$.

*Remark* 5.5. Given the differentiable property of the noisy PS-QNN objective function $\tilde{f}_q(\boldsymbol{\theta})$ (Fontana et al., 2025), we can consider it as a sample drawn from a Gaussian process with the Matern prior covariance function $k_{\mathrm{Matern}-\nu}(\boldsymbol{\theta}, \boldsymbol{\theta}')$ (Appendix A.1), as this Gaussian process allows us to model high-order differentiable functions (Williams & Rasmussen, 2006; Kanagawa et al., 2018).

## 5.2 EFFECTIVE PARAMETER DIMENSION OF THE NOISY PS-QNN

Assuming that Assumption 5.3 holds and using the result of Lemma 5.4, we establish a theoretical limit on the *effective parameter dimension* (Definition 4.1) of the noisy PS-QNNs.

**Theorem 5.6** (Informal). *Consider the Maximum Cut problem on an unweighted $d$-regular graph with $n$ vertices, where $d$ is a constant. Given a constant threshold $\epsilon$ and a noisy PS-QNN $\mathcal{U}_q(\boldsymbol{\theta})$ with $q$-strength local Pauli channels (Eq. 18) that satisfies Assumption 5.3, run BO for $T = \mathrm{poly}(n^{1/\epsilon^2})$ steps, where a predefined scaling parameter $\eta_t$ for the acquisition function $\mathrm{UCB}_t(\boldsymbol{\theta})$ (Eq. 9) is used in each step $t$. Under the condition where the strength $q$ spans $1/\mathrm{poly}(n)$ to $1/n^{1/\sqrt{\log n}}$, if the parameter dimension*

$$p \leq \mathcal{O}\left( \log n / \log(1/q) \right), \tag{22}$$

*then the optimization error $\tilde{r}_T$ (Eq. 20) satisfies $\tilde{r}_T \leq \epsilon$ with high success probability.*

The formal statement and corresponding proof details of Theorem 5.6 are provided in Appendix E. Following our previous perspective of the Bayesian approach, we consider the $n$-qubit noisy PS-QNN objective function $\tilde{f}_q(\boldsymbol{\theta})$ as a sample drawn from a Gaussian process with the Matern covariance function $k_{\mathrm{Matern}-\nu}(\boldsymbol{\theta}, \boldsymbol{\theta}')$. Using this framework, we investigate the trainability of the corresponding noisy PS-QNN $\mathcal{U}_q(\boldsymbol{\theta})$ within a practical scenario concerning the Maximum Cut problem on an unweighted regular graph. Based on Assumption 5.3, we show that if each quantum gate is affected by a $q$-strength local Pauli channel with the strength range of $1/\mathrm{poly}(n)$ to $1/n^{1/\sqrt{\log n}}$, $\mathcal{U}_q(\boldsymbol{\theta})$ with a parameter dimension $p$ of $\mathcal{O}\left( \log n / \log(1/q) \right)$ can also be trained efficiently. For a more intuitive description of the strength range described above, we focus on near-term quantum devices with 50-100 qubits (Preskill, 2018). In this case, $1/n^{1/\sqrt{\log n}}$ is only slightly larger than 0.1. This suggests that this range corresponds to the actual noise levels in near-term quantum devices and holds practical significance.

## 6 CONCLUSION

In this paper, we provide theoretical guarantees regarding the convergence performance of PS-QNNs. We adopt a novel Bayesian approach that considers the PS-QNN objective function as a sample drawn from a specific Gaussian process. By this paradigm shift, we eliminate the need for explicit assumptions about the strong convexity landscape. This enables us to investigate the convergence performance of PS-QNNs in more realistic scenarios, addressing the question of the depth range for efficiently trainable PS-QNNs, as well as analyzing the impact of local Pauli channels on the training of PS-QNNs. Our results shed light on the performance of the QNN and are essential for evaluating its potential quantum advantages in the NISQ era.

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

## A    RELATED DEFINITIONS

This section presents background information on the Matern covariance function, differential entropy, and information gain.

### A.1    MATERN COVARIANCE FUNCTION

The Matern covariance function, widely used in BO, is defined as

$$k_{\text{Matern}-\nu}(\boldsymbol{\theta}, \boldsymbol{\theta}') = \frac{1}{\Gamma(\nu)2^{\nu-1}} \left( \frac{\sqrt{2\nu}d}{l} \right)^{\nu} B_{\nu} \left( \frac{\sqrt{2\nu}d}{l} \right), \tag{23}$$

where $l > 0$, $d = \|\boldsymbol{\theta} - \boldsymbol{\theta}'\|_2$ represents the Euclidean distance between $\boldsymbol{\theta}$ and $\boldsymbol{\theta}'$, $\nu > 0$ denotes the smoothness parameter, $\Gamma(\cdot)$ represents the gamma function, and $B_{\nu}(\cdot)$ denotes the modified Bessel function of the second kind. Varying $\nu$ determines the smoothness of samples drawn from a Gaussian process with this covariance function. Smaller values of $\nu$ correspond to rougher samples. Additionally, these samples are $\lceil \nu \rceil - 1$ times continuously differentiable (Williams & Rasmussen, 2006). Figure 3 illustrates samples drawn from a Gaussian process with this covariance function using different values of $\nu$.

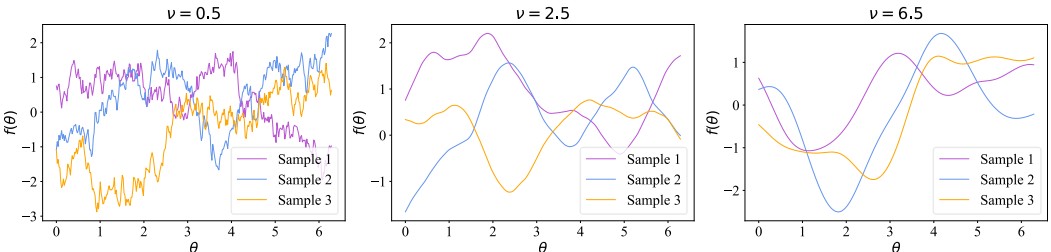

Figure 3: Samples drawn from a Gaussian process with the Matern covariance function $k_{\text{Matern}-\nu}(\boldsymbol{\theta}, \boldsymbol{\theta}')$ using smoothness parameters $\nu$ from $\nu = 0.5$ to $\nu = 6.5$.

### A.2    DIFFERENTIAL ENTROPY

Let $X$ be a random variable with a probability density function $q$ whose support is a set $\mathcal{X}$. The differential entropy $H(X)$ is defined as

$$\mathrm{H}[X] = \mathbb{E}[-\log(q(X))] = \int_{\mathcal{X}} q(x) \log q(x) dx. \tag{24}$$

Specifically, the differential entropy of a multivariate Gaussian random variable $X_{\text{Gaussian}}$ with distribution $N(\boldsymbol{\mu}, \boldsymbol{K})$ is expressed as

$$\mathrm{H}[X_{\text{Gaussian}}] = \frac{1}{2} \log(\det(2\pi e \boldsymbol{K})), \tag{25}$$

where $\boldsymbol{\mu}$ denotes the mean vector and $\boldsymbol{K}$ represents the covariance matrix.

### A.3    INFORMATION GAIN

Let $\mathcal{S}_T = \{(\boldsymbol{\theta}_1, y(\boldsymbol{\theta}_1)), \cdots, (\boldsymbol{\theta}_T, y(\boldsymbol{\theta}_T))\}$ be $T$ accumulated observations about the function $f(\boldsymbol{\theta})$, where $y(\boldsymbol{\theta}_t)$ denotes the estimation of $f(\boldsymbol{\theta}_t)$ for $t \in [T]$. The informativeness of $\mathcal{S}_T$ regarding $f(\boldsymbol{\theta})$ is quantified by the information gain $g_T$, which is the mutual information (Shannon, 1948) between $\boldsymbol{y}_T = [y(\boldsymbol{\theta}_1) \cdots y(\boldsymbol{\theta}_T)]^{\mathsf{T}}$ and $\boldsymbol{f}_T = [f(\boldsymbol{\theta}_1) \cdots f(\boldsymbol{\theta}_T)]^{\mathsf{T}}$. Specifically,

$$g_T = \mathrm{H}[\boldsymbol{y}_T] - \mathrm{H}[\boldsymbol{y}_T | \boldsymbol{f}_T], \tag{26}$$

where $\mathrm{H}[\boldsymbol{y}_T]$ represents the information entropy of $\boldsymbol{y}_T$ and $\mathrm{H}[\boldsymbol{y}_T | \boldsymbol{f}_T]$ denotes the conditional information entropy of $\boldsymbol{y}_T$ given $\boldsymbol{f}_T$.

## B PROOF OF LEMMA 4.3

In this section, we present a complete proof of Lemma 4.3 through a sequence of lemmas. We initially establish the following result regarding the partial derivative $\partial_j f(\boldsymbol{\theta})$ of the noise-free PS-QNN objective function $f(\boldsymbol{\theta}) : \mathcal{D} = [0, 2\pi]^{2p} \mapsto \mathbb{R}$ for any $j \in [2p]$ and any $\boldsymbol{\theta} \in \mathcal{D}$.

**Lemma B.1.** *Assuming that Assumption 4.2 holds, let $f(\boldsymbol{\theta}) : \mathcal{D} = [0, 2\pi]^{2p} \mapsto \mathbb{R}$ be the noise-free PS-QNN objective function. Given a failure probability $\delta \in (0, 1)$, the partial derivative $\partial_j f(\boldsymbol{\theta})$ satisfies*

$$\forall j \in [2p], \forall \boldsymbol{\theta} \in \mathcal{D}, \ |\partial_j f(\boldsymbol{\theta})| \leq \sqrt{\mathbb{V}_{\boldsymbol{\theta}}[\partial_a f(\boldsymbol{\theta})]/\delta} \tag{27}$$

*with a success probability of at least $\geq 1 - \delta$, where $\mathbb{V}_{\boldsymbol{\theta}}[\partial_a f(\boldsymbol{\theta})]$ is the variance of $\partial_a f(\boldsymbol{\theta})$ with index $a = \arg\max_{j \in [2p]}(\sup_{\boldsymbol{\theta} \in \mathcal{D}} |\partial_j f(\boldsymbol{\theta})|)$.*

*Proof.* Fix $a \in [2p]$, by Chebyshev's Inequality, we have

$$\Pr\{\forall \boldsymbol{\theta} \in \mathcal{D}, \forall s > 0, \ |\partial_a f(\boldsymbol{\theta}) - \mathbb{E}_{\boldsymbol{\theta}}[\partial_a f(\boldsymbol{\theta})]| \leq s\} \geq 1 - \mathbb{V}_{\boldsymbol{\theta}}[\partial_a f(\boldsymbol{\theta})]/s^2, \tag{28}$$

where $\mathbb{E}_{\boldsymbol{\theta}}[\partial_a f(\boldsymbol{\theta})]$ and $\mathbb{V}_{\boldsymbol{\theta}}[\partial_a f(\boldsymbol{\theta})]$ are the expectation value and the variance of $\partial_a f(\boldsymbol{\theta})$. Assuming that Assumption 4.2 holds, we demonstrate that $\mathbb{E}_{\boldsymbol{\theta}}[\partial_a f(\boldsymbol{\theta})] = 0$. The detailed proof can be found in Ref. Cerezo et al. (2021b). This implies

$$\Pr\{\forall \boldsymbol{\theta} \in \mathcal{D}, \forall s > 0, \ |\partial_a f(\boldsymbol{\theta})| \leq s\} \geq 1 - \mathbb{V}_{\boldsymbol{\theta}}[\partial_a f(\boldsymbol{\theta})]/s^2. \tag{29}$$

By choosing $a = \arg\max_{j \in [2p]}(\sup_{\boldsymbol{\theta} \in \mathcal{D}} |\partial_j f(\boldsymbol{\theta})|)$, we have

$$\Pr\left\{\forall s > 0, \ \sup_{\boldsymbol{\theta} \in \mathcal{D}} |\partial_a f(\boldsymbol{\theta})| \leq s\right\} \geq 1 - \mathbb{V}_{\boldsymbol{\theta}}[\partial_a f(\boldsymbol{\theta})]/s^2. \tag{30}$$

The use of the index $a$ and the notation $\sup(\cdot)$ immediately implies

$$\Pr\{\forall j \in [2p], \forall \boldsymbol{\theta} \in \mathcal{D}, \forall s > 0, \ |\partial_j f(\boldsymbol{\theta})| \leq s\} \geq 1 - \mathbb{V}_{\boldsymbol{\theta}}[\partial_a f(\boldsymbol{\theta})]/s^2. \tag{31}$$

Let the failure probability $\delta = \mathbb{V}_{\boldsymbol{\theta}}[\partial_a f(\boldsymbol{\theta})]/s^2 \in (0, 1)$, we have

$$\Pr\left\{\forall j \in [2p], \forall \boldsymbol{\theta} \in \mathcal{D}, \ |\partial_j f(\boldsymbol{\theta})| \leq \sqrt{\mathbb{V}_{\boldsymbol{\theta}}[\partial_a f(\boldsymbol{\theta})]/\delta}\right\} \geq 1 - \delta. \tag{32}$$

$\square$

**Lemma B.2.** *Given a noise-free PS-QNN objective function $f(\boldsymbol{\theta}) : \mathcal{D} = [0, 2\pi]^{2p} \mapsto \mathbb{R}$, we have*

$$\forall \boldsymbol{\theta}, \boldsymbol{\theta}' \in \mathcal{D}, \ |f(\boldsymbol{\theta}) - f(\boldsymbol{\theta}')| \leq \max_{j \in [2p]} \left(\sup_{\boldsymbol{\theta} \in \mathcal{D}} |\partial_j f(\boldsymbol{\theta})|\right) \|\boldsymbol{\theta} - \boldsymbol{\theta}'\|_1, \tag{33}$$

*where $\partial_j f(\boldsymbol{\theta})$ is the partial derivative of $f(\boldsymbol{\theta})$ for $j \in [2p]$.*

*Proof.* Let $\boldsymbol{\theta}$ be represented as $[\theta_1, \cdots, \theta_{2p}]^{\mathsf{T}}$. For any $\boldsymbol{\theta}, \boldsymbol{\theta}' \in \mathcal{D}$, we have

$$\begin{aligned} f(\boldsymbol{\theta}) - f(\boldsymbol{\theta}') = &f(\theta_1, \cdots, \theta_{2p}) - f(\theta_1', \theta_2, \cdots, \theta_{2p}) + \cdots + \\ &f(\theta_1', \cdots, \theta_{j-1}', \theta_j, \cdots, \theta_{2p}) - f(\theta_1', \cdots, \theta_j', \theta_{j+1}, \cdots, \theta_{2p}) + \cdots + \\ &f(\theta_1', \cdots, \theta_{2p-1}', \theta_{2p}) - f(\theta_1', \cdots, \theta_{2p}'). \end{aligned} \tag{34}$$

By Triangle Inequality, for any $\boldsymbol{\theta}, \boldsymbol{\theta}' \in \mathcal{D}$, we have

$$\begin{aligned} |f(\boldsymbol{\theta}) - f(\boldsymbol{\theta}')| \leq &|f(\theta_1, \cdots, \theta_{2p}) - f(\theta_1', \theta_2, \cdots, \theta_{2p})| + \cdots + \\ &\left|f(\theta_1', \cdots, \theta_{j-1}', \theta_j, \cdots, \theta_{2p}) - f(\theta_1', \cdots, \theta_j', \theta_{j+1}, \cdots, \theta_{2p})\right| + \cdots + \\ &\left|f(\theta_1', \cdots, \theta_{2p-1}', \theta_{2p}) - f(\theta_1', \cdots, \theta_{2p}')\right|. \end{aligned} \tag{35}$$

For any $j \in [2p]$, the partial derivative with respect to the problem-oriented Hamiltonian $H_1$

$$\partial_j f(\boldsymbol{\theta}) = i\langle\varphi_0|U_-^{\dagger}[H_1, U_+^{\dagger} H_1 U_+]U_-|\varphi_0\rangle \tag{36}$$

and the partial derivative with respect to the mixing Hamiltonian $H_2$

$$\partial_j f(\boldsymbol{\theta}) = i\langle\varphi_0|U_-^\dagger[H_2, U_+^\dagger H_1 U_+]U_-|\varphi_0\rangle \tag{37}$$

exist and are continuous on $\mathcal{D} = [0, 2\pi]^{2p}$, where $U_-$ is the left slice circuit and $U_+$ is the right slice circuit of the variational parameter $\theta_j$ in the noise-free PS-QNN $U(\boldsymbol{\theta})$, and $|\varphi_0\rangle$ is the initial state. Fix $[\theta_1', \cdots, \theta_{j-1}', \theta_{j+1}, \cdots, \theta_{2p}]^\mathsf{T} \in [0, 2\pi]^{2p-1}$, $f(\boldsymbol{\theta})$ can be seen as an uni-variable function in $\theta_j$. By Lagrange's Mean Value Theorem (Sohrab, 2003), for any $\theta_j, \theta_j' \in [0, 2\pi]$ and for any $[\theta_1', \cdots, \theta_{j-1}', \theta_{j+1}, \cdots, \theta_{2p}]^\mathsf{T} \in [0, 2\pi]^{2p-1}$ denoted as $\hat{\boldsymbol{\theta}} \in \hat{\mathcal{D}}$, we have

$$\left|f(\theta_1', \cdots, \theta_{j-1}', \theta_j, \cdots, \theta_{2p}) - f(\theta_1', \cdots, \theta_j', \theta_{j+1}, \cdots, \theta_{2p})\right| \le L_{j,\hat{\boldsymbol{\theta}}}\left|\theta_j - \theta_j'\right|, \tag{38}$$

where $L_{j,\hat{\boldsymbol{\theta}}} = \sup_{\theta_j \in [0, 2\pi]} |\partial_j f(\boldsymbol{\theta})|$. In light of this, for any $\theta_j, \theta_j' \in [0, 2\pi]$ and for any $\hat{\boldsymbol{\theta}} \in \hat{\mathcal{D}}$, we have

$$\left|f(\theta_1', \cdots, \theta_{j-1}', \theta_j, \cdots, \theta_{2p}) - f(\theta_1', \cdots, \theta_j', \theta_{j+1}, \cdots, \theta_{2p})\right| \le L_j\left|\theta_j - \theta_j'\right|, \tag{39}$$

where $L_j = \sup_{\hat{\boldsymbol{\theta}} \in \hat{\mathcal{D}}} L_{j,\hat{\boldsymbol{\theta}}}$. Therefore, for any $\boldsymbol{\theta}, \boldsymbol{\theta}' \in \mathcal{D}$, we have

$$|f(\boldsymbol{\theta}) - f(\boldsymbol{\theta}')| \le L_1\left|\theta_1 - \theta_1'\right| + \cdots + L_{2p}\left|\theta_{2p} - \theta_{2p}'\right| \tag{40}$$

$$\le \left(\max_{j \in [2p]} L_j\right)\sum_{j=1}^{2p}\left|\theta_j - \theta_j'\right| \tag{41}$$

$$= \max_{j \in [2p]} L_j\|\boldsymbol{\theta} - \boldsymbol{\theta}'\|_1 \tag{42}$$

$$= \max_{j \in [2p]}\left(\sup_{\boldsymbol{\theta} \in \mathcal{D}}|\partial_j f(\boldsymbol{\theta})|\right)\|\boldsymbol{\theta} - \boldsymbol{\theta}'\|_1. \tag{43}$$

$\square$

Given Lemma B.1 and Lemma B.2, we come to Lemma 4.3 straightforwardly.

*Proof of Lemma 4.3.* By Lemma B.1, we pick $\delta \in (0, 1)$ and have

$$\Pr\left\{\max_{j \in [2p]}\left(\sup_{\boldsymbol{\theta} \in \mathcal{D}}|\partial_j f(\boldsymbol{\theta})|\right) \le \sqrt{\mathbb{V}_{\boldsymbol{\theta}}[\partial_a f(\boldsymbol{\theta})]/\delta}\right\} \ge 1 - \delta, \tag{44}$$

where $\mathbb{V}_{\boldsymbol{\theta}}[\partial_a f(\boldsymbol{\theta})]$ is the variance of the partial derivative $\partial_a f(\boldsymbol{\theta})$ with index $a = \arg\max_{j \in [2p]}(\sup_{\boldsymbol{\theta} \in \mathcal{D}}|\partial_j f(\boldsymbol{\theta})|)$. Substituting this into Lemma B.2, the statement holds. $\square$

## C   PROOF OF THEOREM 4.5

**Theorem C.1** (Formal). *Given a constant threshold $\epsilon$, a failure probability $\delta \in (0, 1)$ and an $n$-qubit noise-free PS-QNN objective function $f(\boldsymbol{\theta}) : \mathcal{D} = [0, 2\pi]^{2p} \mapsto \mathbb{R}$ induced by the network $U(\boldsymbol{\theta})$ that satisfies Assumption 4.2, run BO for $T = \mathrm{poly}(n^{1/\epsilon^2})$ steps, where the scaling parameter $\eta_t$ for the acquisition function $\mathrm{UCB}_t(\boldsymbol{\theta})$ used in each step $t$ is predefined as*

$$\eta_t = 2\log(2\pi^2 t^2/3\delta) + 4p\log(8\pi pt^2\sqrt{\mathbb{V}_{\boldsymbol{\theta}}[\partial_a f(\boldsymbol{\theta})]/\delta}). \tag{45}$$

*If the parameter dimension*

$$p \le \tilde{\mathcal{O}}\left(\sqrt{\log n}\right), \tag{46}$$

*then the optimization error $r_T$ satisfies $r_T \le \epsilon$ with a success probability of at least $1 - \delta$. Here, $\mathbb{V}_{\boldsymbol{\theta}}[\partial_a f(\boldsymbol{\theta})]$ is the variance of the partial derivative $\partial_a f(\boldsymbol{\theta})$ with index $a = \arg\max_{j \in [2p]}(\sup_{\boldsymbol{\theta} \in \mathcal{D}}|\partial_j f(\boldsymbol{\theta})|)$.*

## C.1 OUTLINE OF THE PROOF PROCEDURE

Our objective is to determine the effective parameter dimension $p$ of the noise-free PS-QNN $U(\boldsymbol{\theta})$ such that the optimization error $r_T = f(\boldsymbol{\theta}^*) - f(\boldsymbol{\theta}_T^+)$ after $T = \text{poly}(n)$ steps of executing BO can be upper bounded by a constant threshold $\epsilon$. Here, $\boldsymbol{\theta}^*$ represents the global maximum point and $\boldsymbol{\theta}_T^+$ denotes the approximation of the maximum point in the previous $T$ steps. We investigate this question through the perspective of the Bayesian approach, which considers the corresponding noise-free PS-QNN objective function $f(\boldsymbol{\theta})$ as a sample drawn from a Gaussian process with the Matern covariance function $k_{\text{Matern}-\nu}(\boldsymbol{\theta}, \boldsymbol{\theta}')$ (Eq. 23). We first establish that $r_T$ is upper bounded by $\frac{1}{T} \sum_{t=1}^{T} (f(\boldsymbol{\theta}^*) - f(\boldsymbol{\theta}_t))$, where $\boldsymbol{\theta}_t$ represents the next point selected in each step $t$. It is evident that the condition $\frac{1}{T} \sum_{t=1}^{T} (f(\boldsymbol{\theta}^*) - f(\boldsymbol{\theta}_t)) \leq \epsilon$ is sufficient to deduce the result $r_T \leq \epsilon$. Hence, by ensuring that the upper bound on $\frac{1}{T} \sum_{t=1}^{T} (f(\boldsymbol{\theta}^*) - f(\boldsymbol{\theta}_t))$ is no greater than $\epsilon$, we can determine the effective $p$ that guarantees $r_T \leq \epsilon$. Subsequently, we utilize the continuity property of the noise-free PS-QNN objective function $f(\boldsymbol{\theta})$ (Lemma 4.3) to establish an upper bound on $\frac{1}{T} \sum_{t=1}^{T} (f(\boldsymbol{\theta}^*) - f(\boldsymbol{\theta}_t))$.

The complete proof of Theorem 4.5 is supported by a series of lemmas (Lemma C.2-Lemma C.8). We will introduce how these lemmas are employed in our proof. For convenience, we initially present explanations of several notions that commonly occur in the following sections. Specifically, $\mathbb{V}_{\boldsymbol{\theta}}[\partial_a f(\boldsymbol{\theta})]$ denotes the variance of the partial derivative $\partial_a f(\boldsymbol{\theta})$ with index $a = \arg\max_{j \in [2p]}(\sup_{\boldsymbol{\theta} \in \mathcal{D}} |\partial_j f(\boldsymbol{\theta})|)$. Additionally, $\mu_{t-1}(\boldsymbol{\theta})$ represents the posterior mean function of $f(\boldsymbol{\theta})$ and $\sigma_{t-1}(\boldsymbol{\theta})$ denotes the posterior standard deviation of $f(\boldsymbol{\theta})$ based on the accumulated observations $\mathcal{S}_{t-1}$ from the previous $t - 1$ steps.

To facilitate the analysis in the continuous domain $\mathcal{D} = [0, 2\pi]^{2p}$, we discretize $\mathcal{D}$ into a finite grid $\mathcal{D}_t$ in each step $t$, as it has been employed in Ref. Srinivas et al. (2012). Specifically, the size of $\mathcal{D}_t$ is determined by the degree of discretization $\tau_t$, such that $|\mathcal{D}_t| = (\tau_t)^{2p}$. In the subsequent discussion, we use $[\boldsymbol{\theta}^*]_t$ to denote the closest point in $\mathcal{D}_t$ to $\boldsymbol{\theta}^*$. Next, we will evaluate upper bounds on $f(\boldsymbol{\theta}^*) - f([\boldsymbol{\theta}^*]_t)$ (the first term) and $f([\boldsymbol{\theta}^*]_t)$ (the second term) to obtain an upper bound on $f(\boldsymbol{\theta}^*)$. Regarding the first term, according to Lemma C.2, if $\tau_t = 8\pi p t^2 \sqrt{\mathbb{V}[\partial_a f(\boldsymbol{\theta})]/\delta}$, then $f(\boldsymbol{\theta}^*) - f([\boldsymbol{\theta}^*]_t)$ can be upper bounded by $1/t^2$ with a success probability of at least $1 - \delta/4$. Considering that $\boldsymbol{\theta}_t$ is selected by maximizing the acquisition function $\text{UCB}_t(\boldsymbol{\theta})$ over $\mathcal{D}$, according to Lemma C.3, $\text{UCB}_t(\boldsymbol{\theta}_t) = \mu_{t-1}(\boldsymbol{\theta}_t) + \sqrt{\eta_t}\sigma_{t-1}(\boldsymbol{\theta}_t)$ can be used to upper bound $f([\boldsymbol{\theta}^*]_t)$ with a success probability of at least $1 - \delta/4$. Here, a predefined scaling parameter $\eta_t = 2\log\left(2\pi^2 t^2 |\mathcal{D}_t|/3\delta\right)$ is used. Taking the two upper bounds mentioned above into account, Lemma C.4 demonstrates that

$$f(\boldsymbol{\theta}^*) = (f(\boldsymbol{\theta}^*) - f([\boldsymbol{\theta}^*]_t)) + f([\boldsymbol{\theta}^*]_t) \leq 1/t^2 + \mu_{t-1}(\boldsymbol{\theta}_t) + \sqrt{\eta_t}\sigma_{t-1}(\boldsymbol{\theta}_t)$$

with a success probability of at least $1 - \delta/2$. Furthermore, we establish that $f(\boldsymbol{\theta}_t)$ is lower bounded by $\mu_{t-1}(\boldsymbol{\theta}_t) - \sqrt{\eta'_t}\sigma_{t-1}(\boldsymbol{\theta}_t)$ with a success probability of at least $1 - \delta/2$ using Lemma C.5, where $\eta'_t = 2\log(\pi^2 t^2/3\delta)$. Since $\eta_t \geq \eta'_t$, we can also use $\mu_{t-1}(\boldsymbol{\theta}_t) - \sqrt{\eta_t}\sigma_{t-1}(\boldsymbol{\theta}_t)$ as a lower bound for $f(\boldsymbol{\theta}_t)$. Afterward, Lemma C.6 establishes that

$$f(\boldsymbol{\theta}^*) - f(\boldsymbol{\theta}_t) \leq 1/t^2 + 2\sqrt{\eta_t}\sigma_{t-1}(\boldsymbol{\theta}_t)$$

with a success probability of at least $1 - \delta$. Then, Lemma C.7 establishes a connection between the sum of posterior variances $\sum_{t=1}^{T} \sigma_{t-1}^2(\boldsymbol{\theta}_t)$ and the information gain $g_T$ (Eq. 26). As $f(\boldsymbol{\theta})$ is considered as a sample drawn from a Gaussian process with $k_{\text{Matern}-\nu}(\boldsymbol{\theta}, \boldsymbol{\theta}')$, we can bound $\sum_{t=1}^{T} \sigma_{t-1}^2(\boldsymbol{\theta}_t)$ by the upper bound $\mathcal{O}(T^{\frac{p}{v+p}} \log^{\frac{v}{v+p}}(T))$ on the maximal $g_T$ for $k_{\text{Matern}-\nu}(\boldsymbol{\theta}, \boldsymbol{\theta}')$ in Ref. Vakili et al. (2021). By applying Cauchy-Schwarz Inequality and considering the non-decreasing property of $\eta_t$ as $t$ increases, we can substitute the form of $\eta_T$ to obtain the result stated in Lemma C.8

$$r_T \leq \mathcal{O}\left(\sqrt{p \log\left(pT^2(\mathbb{V}_{\boldsymbol{\theta}}[\partial_a f(\boldsymbol{\theta})])^{1/2}\right)} (\log T/T)^{\frac{\nu}{\nu+p}}\right)$$

with a success probability of at least $1 - \delta$. Finally, we obtain the effective $p$ by solving for this upper bound is no greater than a constant threshold $\epsilon$ with $T = \text{poly}(n^{1/\epsilon^2})$.

## C.2 PROOF DETAILS

In this section, we provide a comprehensive introduction to the corresponding lemmas.

**Lemma C.2.** *Assuming that Assumption 4.2 holds, let $f(\boldsymbol{\theta}) : \mathcal{D} = [0, 2\pi]^{2p} \mapsto \mathbb{R}$ be the $n$-qubit noise-free PS-QNN objective function. Given a failure probability $\delta \in (0, 1)$ and a finite grid $\mathcal{D}_t$ of size $|\mathcal{D}_t| = (\tau_t)^{2p}$ with the degree of discretization $\tau_t = 4\pi p t^2 \sqrt{\mathbb{V}[\partial_a f(\boldsymbol{\theta})]/\delta}$ in each step $t$, run BO for $T = \mathrm{poly}(n)$ steps. The following relationship*

$$\forall t \in [T], \forall \boldsymbol{\theta} \in \mathcal{D}, \ |f(\boldsymbol{\theta}) - f([\boldsymbol{\theta}]_t)| \leq 1/t^2 \tag{47}$$

*holds with a success probability of at least $1 - \delta$, where $[\boldsymbol{\theta}]_t$ represents the closest point in $\mathcal{D}_t$ to $\boldsymbol{\theta}$.*

*Proof.* By choosing a finite grid $\mathcal{D}_t$ of size $(\tau_t)^{2p}$ in each step $t$, for any $\boldsymbol{\theta} \in \mathcal{D}$ we have $\|\boldsymbol{\theta} - [\boldsymbol{\theta}]_t\|_1 \leq 4\pi p/\tau_t$. Given Lemma 4.3, we have

$$\Pr\left\{\forall t \in [T], \forall \boldsymbol{\theta} \in \mathcal{D}, \ |f(\boldsymbol{\theta}) - f([\boldsymbol{\theta}]_t)| \leq 4\pi p \sqrt{\mathbb{V}[\partial_a f(\boldsymbol{\theta})]/\delta}/\tau_t\right\} \geq 1 - \delta, \tag{48}$$

where the failure probability $\delta \in (0, 1)$. Since $\tau_t = 4\pi p t^2 \sqrt{\mathbb{V}[\partial_a f(\boldsymbol{\theta})]/\delta}$, then

$$\Pr\left\{\forall t \in [T], \forall \boldsymbol{\theta} \in \mathcal{D}, \ |f(\boldsymbol{\theta}) - f([\boldsymbol{\theta}]_t)| \leq 1/t^2\right\} \geq 1 - \delta. \tag{49}$$

Furthermore, we consider $\mathbb{V}[\partial_a f(\boldsymbol{\theta})]$ to be $1/\mathrm{poly}(n)$, as shown in Ref. Park & Killoran (2024). Additionally, we assume that parameter dimension $p$ is at most $\mathrm{poly}(n)$. In order to guarantee the degree of discretization $\tau_t$ of at least 1, we enforce a constraint that the number of steps $T = \mathrm{poly}(n)$. This constraint is consistent with the scenario we are exploring. $\square$

**Lemma C.3.** *Given a failure probability $\delta \in (0, 1)$, an $n$-qubit noise-free PS-QNN objective function $f(\boldsymbol{\theta}) : \mathcal{D} = [0, 2\pi]^{2p} \mapsto \mathbb{R}$ and a finite grid $\mathcal{D}_t \subset \mathcal{D}$ of size $|\mathcal{D}_t|$ in each step $t$, run BO for $T = \mathrm{poly}(n)$ steps, where a scaling parameter $\eta_t$ for the acquisition function $\mathrm{UCB}_t(\boldsymbol{\theta})$ used in each step $t$ is predefined as $\eta_t = 2\log(\pi^2 t^2 |\mathcal{D}_t|/6\delta)$. The following relationship*

$$\forall t \in [T], \forall \boldsymbol{\theta} \in \mathcal{D}_t, \ f(\boldsymbol{\theta}) \in \mathcal{C}_t(\boldsymbol{\theta}) \tag{50}$$

*holds with a success probability of at least $1 - \delta$, where $\mathcal{C}_t(\boldsymbol{\theta})$ represents a confidence interval $[\mu_{t-1}(\boldsymbol{\theta}) - \sqrt{\eta_t}\sigma_{t-1}(\boldsymbol{\theta}), \ \mu_{t-1}(\boldsymbol{\theta}) + \sqrt{\eta_t}\sigma_{t-1}(\boldsymbol{\theta})]$.*

*Proof.* Fix $t \in [T]$ and $\boldsymbol{\theta} \in \mathcal{D}_t$. Conditioned on accumulated observations $\mathcal{S}_{t-1}$ from the previous $t - 1$ steps, the posterior distribution $f(\boldsymbol{\theta}) \sim N(\mu_{t-1}(\boldsymbol{\theta}), \sigma_{t-1}^2(\boldsymbol{\theta}))$. Now, if $b \sim N(0, 1)$, then

$$\Pr\{b > w\} = \exp(-w^2/2)(2\pi)^{-1/2} \exp\left(-(b-w)^2/2 - w(b-w)\right) \tag{51}$$

$$\leq \exp(-w^2/2) \Pr\{b > 0\} \tag{52}$$

$$= \frac{1}{2}\exp(-w^2/2) \tag{53}$$

for $w > 0$, since $\exp(-w(b-w)) \leq 1$ for $b \geq w$. Using $b = (f(\boldsymbol{\theta}) - \mu_{t-1}(\boldsymbol{\theta}))/\sigma_{t-1}(\boldsymbol{\theta})$ and $w = \sqrt{\eta_t}$, we have

$$\Pr\{f(\boldsymbol{\theta}) \notin \mathcal{C}_t(\boldsymbol{\theta})\} \leq \exp(-\eta_t/2). \tag{54}$$

Applying the union bound for $\boldsymbol{\theta} \in \mathcal{D}_t$, we have

$$\Pr\{\forall \boldsymbol{\theta} \in \mathcal{D}_t, \ f(\boldsymbol{\theta}) \in \mathcal{C}_t(\boldsymbol{\theta})\} \geq 1 - |\mathcal{D}_t| \exp(-\eta_t/2). \tag{55}$$

Given that $|\mathcal{D}_t| \exp(-\eta_t/2) = \delta/q_t$, where $\sum_{t \geq 1}(1/q_t) = 1$, $q_t > 0$, by applying the union bound for $t \in \mathbb{N}$, the statement holds. For example, we can use $q_t = \pi^2 t^2/6$. $\square$

**Lemma C.4.** *Assuming that Assumption 4.2 holds, let $f(\boldsymbol{\theta}) : \mathcal{D} = [0, 2\pi]^{2p} \mapsto \mathbb{R}$ be the $n$-qubit noise-free PS-QNN objective function. Given a failure probability $\delta \in (0, 1)$, run BO for $T = \mathrm{poly}(n)$ steps, where a scaling parameter $\eta_t$ for the acquisition function $\mathrm{UCB}_t(\boldsymbol{\theta})$ used in each step $t$ is predefined as $\eta_t = 2\log(\pi^2 t^2/3\delta) + 4p\log(4\pi p t^2 \sqrt{2\mathbb{V}_{\boldsymbol{\theta}}[\partial_a f(\boldsymbol{\theta})]/\delta})$. The following relationship*

$$\forall t \in [T], \ f(\boldsymbol{\theta}^*) \leq \mu_{t-1}(\boldsymbol{\theta}_t) + \sqrt{\eta_t}\sigma_{t-1}(\boldsymbol{\theta}_t) + 1/t^2 \tag{56}$$

*holds with a success probability of at least $1 - \delta$, where $\boldsymbol{\theta}^*$ denotes the global maximum point and $\boldsymbol{\theta}_t$ represents the next point selected in each step $t$.*

*Proof.* Using the failure probability $\delta/2$ in Lemma C.2, for the global maximum point $\boldsymbol{\theta}^*$, we have

$$\Pr\{\forall t \in [T],\ f(\boldsymbol{\theta}^*) - f([\boldsymbol{\theta}^*]_t) \leq 1/t^2\} \geq 1 - \delta/2, \tag{57}$$

where $[\boldsymbol{\theta}^*]_t$ denotes the closest point in $\mathcal{D}_t$ to $\boldsymbol{\theta}^*$. Here, a finite grid $\mathcal{D}_t$ of size $|\mathcal{D}_t| = (\tau_t)^{2p}$ with $\tau_t = 4\pi pt^2 \sqrt{2\mathbb{V}[\partial_a f(\boldsymbol{\theta})]/\delta}$. Then, applying Lemma C.3 with the failure probability $\delta/2$, for $[\boldsymbol{\theta}^*]_t$, we have

$$\Pr\{\forall t \in [T],\ f([\boldsymbol{\theta}^*]_t) \leq \mu_{t-1}([\boldsymbol{\theta}^*]_t) + \sqrt{\eta_t}\sigma_{t-1}([\boldsymbol{\theta}^*]_t)\} \geq 1 - \delta/2, \tag{58}$$

where $\eta_t = 2\log(\pi^2 t^2 |\mathcal{D}_t|/3\delta)$. As the next point $\boldsymbol{\theta}_t$ is selected by maximizing $\mathrm{UCB}_t(\boldsymbol{\theta})$ in each step $t$, we have $\mathrm{UCB}_t([\boldsymbol{\theta}^*]_t) \leq \mathrm{UCB}_t(\boldsymbol{\theta}_t)$. Then, we have

$$\Pr\{\forall t \in [T],\ f([\boldsymbol{\theta}^*]_t) \leq \mu_{t-1}(\boldsymbol{\theta}_t) + \sqrt{\eta_t}\sigma_{t-1}(\boldsymbol{\theta}_t)\} \geq 1 - \delta/2. \tag{59}$$

Taking Eq. 57 and Eq. 59 together, the statement holds since $(1 - \delta/2)^2 > 1 - \delta$. □

**Lemma C.5.** *Given a failure probability $\delta \in (0,1)$ and an $n$-qubit noise-free PS-QNN objective function $f(\boldsymbol{\theta}) : \mathcal{D} = [0, 2\pi]^{2p} \mapsto \mathbb{R}$, run BO for $T = \mathrm{poly}(n)$ steps, where a scaling parameter $\eta_t'$ for the acquisition function $\mathrm{UCB}_t(\boldsymbol{\theta})$ used in each step $t$ is predefined as $\eta_t' = 2\log(\pi^2 t^2/6\delta)$. The following relationship*

$$\forall t \in [T],\ f(\boldsymbol{\theta}_t) \in \mathcal{C}_t(\boldsymbol{\theta}_t) \tag{60}$$

*holds with a success probability of at least $1 - \delta$, where $\boldsymbol{\theta}_t$ represents the next point selected in each step $t$ and $\mathcal{C}_t(\boldsymbol{\theta}_t)$ denotes the confidence interval $[\mu_{t-1}(\boldsymbol{\theta}_t) - \sqrt{\eta_t'}\sigma_{t-1}(\boldsymbol{\theta}_t),\ \mu_{t-1}(\boldsymbol{\theta}_t) + \sqrt{\eta_t'}\sigma_{t-1}(\boldsymbol{\theta}_t)]$.*

*Proof.* Fix $t \in [T]$. Conditioned on $\mathcal{S}_{t-1}$ from the previous $t-1$ steps, for the next point $\boldsymbol{\theta}_t$ selected in each step $t$, the posterior distribution $f(\boldsymbol{\theta}_t) \sim N(\mu_{t-1}(\boldsymbol{\theta}_t), \sigma_{t-1}^2(\boldsymbol{\theta}_t))$. Now, if $b \sim N(0,1)$, then $\Pr\{b > w\} \leq \frac{1}{2}\exp(-w^2/2)$ for $w > 0$. Using $b = (f(\boldsymbol{\theta}_t) - \mu_{t-1}(\boldsymbol{\theta}_t))/\sigma_{t-1}(\boldsymbol{\theta}_t)$ and $w = \sqrt{\eta_t'}$, we have

$$\Pr\{f(\boldsymbol{\theta}_t) \notin \mathcal{C}_t(\boldsymbol{\theta}_t)\} \leq \exp(-\eta_t'/2). \tag{61}$$

Given that $\exp(-\eta_t'/2) = \delta/q_t$, where $\sum_{t \geq 1}(1/q_t) = 1$, $q_t > 0$, by applying the union bound for $t \in \mathbb{N}$, the statement holds. For example, we can use $q_t = \pi^2 t^2/6$. □

**Lemma C.6.** *Assuming that Assumption 4.2 holds, let $f(\boldsymbol{\theta}) : \mathcal{D} = [0, 2\pi]^{2p} \mapsto \mathbb{R}$ be the $n$-qubit noise-free PS-QNN objective function. Given a failure probability $\delta \in (0,1)$, run BO for $T = \mathrm{poly}(n)$ steps, where a scaling parameter $\eta_t$ for the acquisition function $\mathrm{UCB}_t(\boldsymbol{\theta})$ used in each step $t$ is predefined as $\eta_t = 2\log(2\pi^2 t^2/3\delta) + 4p\log(8\pi pt^2\sqrt{\mathbb{V}[\partial_a f(\boldsymbol{\theta})]/\delta})$. The following relationship*

$$\forall t \in [T],\ f(\boldsymbol{\theta}^*) - f(\boldsymbol{\theta}_t) \leq 2\sqrt{\eta_t}\sigma_{t-1}(\boldsymbol{\theta}_t) + 1/t^2 \tag{62}$$

*holds with a success probability of at least $1 - \delta$, where $\boldsymbol{\theta}^*$ denotes the global maximum point and $\boldsymbol{\theta}_t$ represents the next point selected in each step $t$.*

*Proof.* Using the failure probability $\delta/2$ in Lemma C.4, for the global maximum point $\boldsymbol{\theta}^*$, we have

$$\Pr\{\forall t \in [T],\ f(\boldsymbol{\theta}^*) \leq \mu_{t-1}(\boldsymbol{\theta}_t) + \sqrt{\eta_t}\sigma_{t-1}(\boldsymbol{\theta}_t) + 1/t^2\} \geq 1 - \delta/2 \tag{63}$$

with $\eta_t = 2\log(2\pi^2 t^2/3\delta) + 4p\log(8\pi pt^2\sqrt{\mathbb{V}[\partial_a f(\boldsymbol{\theta})]/\delta})$ in each step $t$. Then, using the failure probability $\delta/2$ in Lemma C.5, for the next point $\boldsymbol{\theta}_t$ selected in each step $t$, we have

$$\Pr\{\forall t \in [T],\ f(\boldsymbol{\theta}_t) \geq \mu_{t-1}(\boldsymbol{\theta}_t) - \sqrt{\eta_t'}\sigma_{t-1}(\boldsymbol{\theta}_t)\} \geq 1 - \delta/2 \tag{64}$$

with $\eta_t' = 2\log(\pi^2 t^2/3\delta)$ in each step $t$. As the aforementioned $\eta_t$ is greater than $\eta_t'$ used here, choosing $\eta_t$ here is also valid. Taking Eq. 63 and Eq. 64 together, the proof is completed. □

**Lemma C.7.** *Given an $n$-qubit noise-free PS-QNN objective function $f(\boldsymbol{\theta}) : \mathcal{D} = [0, 2\pi]^{2p} \mapsto \mathbb{R}$, run BO for $T = \mathrm{poly}(n)$ steps. Let $\mathcal{S}_T = \{(\boldsymbol{\theta}_1, y(\boldsymbol{\theta}_1)), \cdots, (\boldsymbol{\theta}_T, y(\boldsymbol{\theta}_T))\}$ be the accumulated observations from the previous $T$ steps, where the estimation $y(\boldsymbol{\theta}_t) = f(\boldsymbol{\theta}_t) + \xi_t^{\mathrm{noise}}$ in each step $t$. Here, $\xi_t^{\mathrm{noise}} \sim N(0, 1/4M)$ is independent and identically distributed Gaussian noise with $M$ representing the fixed number of measurements. The information gain $g_T$ (Eq. 26) can be expressed as*

$$g_T = \frac{1}{2}\sum_{t=1}^{T}\log(1 + 4M\sigma_{t-1}^2(\boldsymbol{\theta}_t)). \tag{65}$$

*Proof.* Let $\boldsymbol{y}_{t-1} = [y(\boldsymbol{\theta}_1) \cdots y(\boldsymbol{\theta}_{t-1})]^\mathsf{T}$ and $\boldsymbol{f}_{t-1} = [f(\boldsymbol{\theta}_1) \cdots f(\boldsymbol{\theta}_{t-1})]^\mathsf{T}$ for $t \in [T+1]$. Plugging in the differential entropy of a multivariate Gaussian random variable (Eq. 25), we have $\mathrm{H}[y(\boldsymbol{\theta}_t)|\boldsymbol{y}_{t-1}] = 1/2\log(2\pi e(1/4M + \sigma_{t-1}^2(\boldsymbol{\theta}_t)))$ for $t \in [T]$ and $\mathrm{H}[\boldsymbol{y}_T|\boldsymbol{f}_T] = \frac{T}{2}\log(\pi e/2M)$. Using the fact that $\mathrm{H}[\boldsymbol{y}_t] = \mathrm{H}[\boldsymbol{y}_{t-1}] + \mathrm{H}[y(\boldsymbol{\theta}_t)|\boldsymbol{y}_{t-1}]$, we have

$$\mathrm{H}[\boldsymbol{y}_T] = \mathrm{H}[\boldsymbol{y}_0] + \mathrm{H}[y(\boldsymbol{\theta}_1)|\boldsymbol{y}_0] + \mathrm{H}[y(\boldsymbol{\theta}_2)|\boldsymbol{y}_1] + \cdots + \mathrm{H}[y(\boldsymbol{\theta}_T)|\boldsymbol{y}_{T-1}] \tag{66}$$

$$= \frac{1}{2}\sum_{t=1}^{T}\log(2\pi e(1/4M + \sigma_{t-1}^2(\boldsymbol{\theta}_t))). \tag{67}$$

Recalling the definition of $g_T$ (Eq. 26), the statement holds. $\qquad\square$

**Lemma C.8.** *Assuming that Assumption 4.2 holds, let $f(\boldsymbol{\theta}) : \mathcal{D} = [0, 2\pi]^{2p} \mapsto \mathbb{R}$ be the n-qubit noise-free PS-QNN objective function. Given a failure probability $\delta \in (0, 1)$, run BO with the Matern prior covariance function $k_{\mathrm{Matern}-\nu}(\boldsymbol{\theta}, \boldsymbol{\theta}')$ (Eq. 23) for $T = \mathrm{poly}(n)$ steps, where a scaling parameter $\eta_t$ for the acquisition function $\mathrm{UCB}_t(\boldsymbol{\theta})$ used in each step $t$ is predefined as $\eta_t = 2\log(2\pi^2 t^2/3\delta) + 4p\log(8\pi pt^2\sqrt{\mathbb{V}_{\boldsymbol{\theta}}[\partial_a f(\boldsymbol{\theta})]/\delta})$. The optimization error $r_T$ satisfies*

$$r_T \leq \mathcal{O}\left(\sqrt{p\log\left(pT^2(\mathbb{V}_{\boldsymbol{\theta}}[\partial_a f(\boldsymbol{\theta})])^{1/2}\right)}(\log T/T)^{\frac{\nu}{\nu+p}}\right) \tag{68}$$

*with a success probability of at least $1 - \delta$.*

*Proof.* Noted that $\eta_t$ in Lemma C.6 is non-decreasing. Since $0 \leq 4M\sigma_{t-1}^2(\boldsymbol{\theta}_t) \leq 4Mk_{\mathrm{Matern}-\nu}(\boldsymbol{\theta}_t, \boldsymbol{\theta}_t) \leq 4M$, denoted as $4M\sigma_{t-1}^2(\boldsymbol{\theta}_t) \in [0, 4M]$, we have $4M\sigma_{t-1}^2(\boldsymbol{\theta}_t) \leq (4M/\log(1 + 4M))\log(1 + 4M\sigma_{t-1}^2(\boldsymbol{\theta}_t))$. Moreover, Lemma C.7 links the sum of the posterior variances $\sum_{t=1}^{T} \sigma_{t-1}^2(\boldsymbol{\theta}_t)$ to the information gain $g_T$. By Cauchy-Schwarz Inequality, we have

$$\left(\sum_{t=1}^{T} 2\sqrt{\eta_t}\sigma_{t-1}(\boldsymbol{\theta}_t)\right)^2 \leq \sum_{t=1}^{T} 4\eta_t \sum_{t=1}^{T} \sigma_{t-1}^2(\boldsymbol{\theta}_t) \tag{69}$$

$$\leq \frac{T\eta_T}{M}\sum_{t=1}^{T}(4M\sigma_{t-1}^2(\boldsymbol{\theta}_t)) \tag{70}$$

$$\leq \frac{4T\eta_T}{\log(1 + 4M)}\sum_{t=1}^{T}\log(1 + 4M\sigma_{t-1}^2(\boldsymbol{\theta}_t)) \tag{71}$$

$$= c_0 T\eta_T g_T, \tag{72}$$

where the parameter $c_0 = 8/\log(1 + 4M)$. The optimization error is given by $r_T = f(\boldsymbol{\theta}^*) - f(\boldsymbol{\theta}_T^+)$, where $\boldsymbol{\theta}^*$ represents the global maximum point and $\boldsymbol{\theta}_T^+ = \arg\max_{\boldsymbol{\theta} \in \mathcal{A}_T} f(\boldsymbol{\theta})$ denotes the approximation of the maximum point with the accumulated points $\mathcal{A}_T = \{\boldsymbol{\theta}_1, \cdots, \boldsymbol{\theta}_T\}$ from the previous $T$ steps. Now, we have

$$r_T \leq \frac{1}{T}\sum_{t=1}^{T}(f(\boldsymbol{\theta}^*) - f(\boldsymbol{\theta}_t)) \tag{73}$$

$$\leq \frac{1}{T}\left(\sum_{t=1}^{T} 2\sqrt{\eta_t}\sigma_{t-1}(\boldsymbol{\theta}_t) + \sum_{t=1}^{T} 1/t^2\right) \tag{74}$$

$$\leq \frac{1}{T}\left(\sqrt{c_0 T\eta_T g_T} + \pi^2/6\right). \tag{75}$$

As $f(\boldsymbol{\theta})$ is considered as a sample drawn from a Gaussian process with $k_{\mathrm{Matern}-\nu}(\boldsymbol{\theta}, \boldsymbol{\theta}')$, we can use the upper bound $\mathcal{O}(T^{\frac{p}{\nu+p}}\log^{\frac{\nu}{\nu+p}}(T))$ on the maximal $g_T$ for $k_{\mathrm{Matern}-\nu}(\boldsymbol{\theta}, \boldsymbol{\theta}')$ in Ref. Vakili et al. (2021). By substituting $\eta_T$ and $\mathcal{O}(T^{\frac{p}{\nu+p}}\log^{\frac{\nu}{\nu+p}}(T))$ into Eq. 75, the statement holds. $\qquad\square$

Now we are ready to complete the proof of Theorem 4.5.

*Proof of Theorem 4.5.* We consider $\mathbb{V}[\partial_a f(\boldsymbol{\theta})]$ to be $1/\text{poly}(n)$, as shown in Ref. Park & Killoran (2024). Additionally, we assume that the parameter dimension $p$ is at most $\text{poly}(n)$. To ensure consistency with the scenario under investigation and to guarantee the degree of discretization $\tau_t$ in Lemma C.2 of at least 1, we impose a constraint that the number of steps $T = \text{poly}(n)$. Hence, it is reasonable to treat $\log\left(pT^2(\mathbb{V}_{\boldsymbol{\theta}}[\partial_a f(\boldsymbol{\theta})])^{1/2}\right)$ as a constant. Therefore, our task is to find the effective $p$ that satisfies the condition $(p(\log(T)/T)^{\frac{\nu}{\nu+p}})^{1/2} \le \epsilon$, where $\epsilon$ is a constant threshold and $T = \text{poly}(n)$. Let

$$p \le \frac{1}{2}\left(\epsilon^2 - \nu + \sqrt{(\epsilon^2 - \nu)^2 + 4\nu\epsilon^2\left(1 + \log(T/\log T)\right)}\right), \tag{76}$$

then the above upper bound satisfies the inequality

$$p^2 - (\epsilon^2 - \nu)p - \nu\epsilon^2\left(1 + \log\left(T/\log T\right)\right) \le 0. \tag{77}$$

Equivalently, the above inequality can be rewritten by

$$\log\left(\log T/T\right) \le \left(1 + p/\nu\right)\left(1 - p/\epsilon^2\right). \tag{78}$$

Considering the relationship $\log x \ge 1 - 1/x$ holds for $x > 0$, then the above inequality implies

$$\log\left(\log T/T\right) \le \left(1 + p/\nu\right)\log\left(\epsilon^2/p\right), \tag{79}$$

which directly leads to

$$\log T/T \le \left(\epsilon^2/p\right)^{1+p/\nu}, \tag{80}$$

that is $(p(\log(T)/T)^{\frac{\nu}{\nu+p}})^{1/2} \le \epsilon$. Finally, let $T = \text{poly}(n^{1/\epsilon^2})$ and substitute it into Eq. 76. We obtain the effective parameter dimension $p$ for the noise-free PS-QNN, which is $p \le \tilde{\mathcal{O}}\left(\sqrt{\log n}\right)$. $\square$

# D PROOF OF LEMMA 5.4

In this section, we provide the proof of Lemma 5.4 which is similar to the proof of Lemma 4.3.

*Proof of Lemma 5.4.* Given an $n$-qubit noisy PS-QNN objective function with $q$-strength local Pauli channels $\tilde{f}_q(\boldsymbol{\theta}) : \mathcal{D} = [0, 2\pi]^{2p} \mapsto \mathbb{R}$, for any $j \in [2p]$, the partial derivatives $\partial_j \tilde{f}_q(\boldsymbol{\theta})$ exist and are continuous, as shown in Ref. Wang et al. (2021). Using a similar proof sketch as in Lemma B.2, we have

$$\forall \boldsymbol{\theta}, \boldsymbol{\theta}' \in \mathcal{D}, \ \left|\tilde{f}_q(\boldsymbol{\theta}) - \tilde{f}_q(\boldsymbol{\theta}')\right| \le \max_{j \in [2p]}\left(\sup_{\boldsymbol{\theta} \in \mathcal{D}}\left|\partial_j \tilde{f}_q(\boldsymbol{\theta})\right|\right)\|\boldsymbol{\theta} - \boldsymbol{\theta}'\|_1. \tag{81}$$

Considering the Maximum Cut problem on an unweighted $d$-regular graph with $n$ vertices, we can rely on Corollary 2 in Ref. Wang et al. (2021) to obtain an upper bound on $\partial_j \tilde{f}_q(\boldsymbol{\theta})$ for any $j \in [2p]$. Then, the following relationship

$$\forall \boldsymbol{\theta}, \boldsymbol{\theta}' \in \mathcal{D}, \ \left|\tilde{f}_q(\boldsymbol{\theta}) - \tilde{f}_q(\boldsymbol{\theta}')\right| \le L\|\boldsymbol{\theta} - \boldsymbol{\theta}'\|_1 \tag{82}$$

holds, where the Lipschitz continuity factor is given by

$$L = \sqrt{\ln 2/2}d^2 n^{\frac{5}{2}}\|H_1^{\text{MaxCut}}\|_\infty q^{((d_1+1)p+1)} \tag{83}$$

with the strength $q \in (0, 1)$ and $d_1$ representing the network depth of the implementation of the unitary corresponding to the problem-oriented Hamiltonian $H_1^{\text{MaxCut}}$. Since $\|H_1^{\text{MaxCut}}\|_\infty = \mathcal{O}(nd/2)$, $q \in (0, 1)$ and $d_1 = \Omega(d)$, we have $L = \mathcal{O}(d^3 n^{7/2} q^{(d+1)p})$. Thus, the proof of Lemma 5.4 is concluded. $\square$

# E  PROOF OF THEOREM 5.6

**Theorem E.1** (Formal). *Consider the Maximum Cut problem on an unweighted $d$-regular graph with $n$ vertices, where $d$ is a constant. Given a constant threshold $\epsilon$, a failure probability $\delta \in (0,1)$ and a noisy PS-QNN objective function with $q$-strength local Pauli channels $\tilde{f}_q(\boldsymbol{\theta})$ : $\mathcal{D} = [0, 2\pi]^{2p} \mapsto \mathbb{R}$ induced by the network $\mathcal{U}_q(\boldsymbol{\theta})$ that satisfies Assumption 5.3, run BO for $T = \mathrm{poly}(n^{1/\epsilon^2})$ steps, where the scaling parameter $\eta_t$ for the acquisition function $\mathrm{UCB}_t(\boldsymbol{\theta})$ used in each step $t$ is predefined as*

$$\eta_t = 2\log(\pi^2 t^2/(3\delta)) + 4p\log(4\pi pt^2 d^3 n^{7/2} q^{(d+1)p}). \tag{84}$$

*Under the condition where the strength $q$ spans $1/\mathrm{poly}(n)$ to $1/n^{1/\sqrt{\log n}}$, if the parameter dimension*

$$p \leq \mathcal{O}\left(\log n / \log(1/q)\right), \tag{85}$$

*then the optimization error $\tilde{r}_T$ satisfies $\tilde{r}_T \leq \epsilon$ with a success probability of at least $1 - \delta$.*

## E.1  OUTLINE OF THE PROOF PROCEDURE

Our objective is to determine the effective parameter dimension $p$ of the noisy PS-QNN $\mathcal{U}_q(\boldsymbol{\theta})$ such that the optimization error $\tilde{r}_T = \tilde{f}_q(\tilde{\boldsymbol{\theta}}^*) - \tilde{f}_q(\tilde{\boldsymbol{\theta}}_T^+)$ after $T = \mathrm{poly}(n)$ steps of executing BO can be upper bounded by a constant threshold $\epsilon$. Here, $\tilde{\boldsymbol{\theta}}^*$ represents the global maximum point and $\tilde{\boldsymbol{\theta}}_T^+$ denotes the approximation of the maximum point in the previous $T$ steps. We investigate this question through the perspective of the Bayesian approach, which considers the corresponding noisy PS-QNN objective function $\tilde{f}_q(\boldsymbol{\theta})$ as a sample drawn from a Gaussian process with the Matern covariance function $k_{\mathrm{Matern}-\nu}(\boldsymbol{\theta}, \boldsymbol{\theta}')$. We first establish that $\tilde{r}_T$ is upper bounded by $\frac{1}{T}\sum_{t=1}^{T}(\tilde{f}_q(\tilde{\boldsymbol{\theta}}^*) - \tilde{f}_q(\tilde{\boldsymbol{\theta}}_t))$, where $\tilde{\boldsymbol{\theta}}_t$ represents the next point selected in each step $t$. It is evident that the condition $\frac{1}{T}\sum_{t=1}^{T}(\tilde{f}_q(\tilde{\boldsymbol{\theta}}^*) - \tilde{f}_q(\tilde{\boldsymbol{\theta}}_t)) \leq \epsilon$ is sufficient to deduce the result $\tilde{r}_T \leq \epsilon$. Hence, by ensuring that the upper bound on $\frac{1}{T}\sum_{t=1}^{T}(\tilde{f}_q(\tilde{\boldsymbol{\theta}}^*) - \tilde{f}_q(\tilde{\boldsymbol{\theta}}_t))$ is no greater than $\epsilon$, we can determine the effective $p$ that guarantees $\tilde{r}_T \leq \epsilon$. Subsequently, we utilize the continuity property of the noisy PS-QNN objective function $\tilde{f}_q(\boldsymbol{\theta})$ (Lemma 5.4) to establish an upper bound on $\frac{1}{T}\sum_{t=1}^{T}(\tilde{f}_q(\tilde{\boldsymbol{\theta}}^*) - \tilde{f}_q(\tilde{\boldsymbol{\theta}}_t))$. The complete proof of Theorem 5.6 is similar to the proof of Theorem 4.5 and is supported by a series of lemmas analogous to Lemma C.2 to Lemma C.8. Instead of providing a detailed description of each lemma here, we will directly present lemma E.2 similar to Lemma C.8. Additionally, we will emphasize the impact of the difference in continuity property between the noise-free and noisy PS-QNN objective functions on the result.

## E.2  PROOF DETAILS

In this section, we provide a comprehensive introduction to the Lemma E.2.

**Lemma E.2.** *Considering a Maximum Cut problem on an unweighted $d$-regular graph with $n$ vertices, where $d$ is a constant. Assuming that Assumption 5.3 holds, let $\tilde{f}_q(\boldsymbol{\theta}) : \mathcal{D} = [0, 2\pi]^{2p} \mapsto \mathbb{R}$ be the noisy PS-QNN objective function with $q$-strength local Pauli channels, where the strength $q \in (0,1)$. Given a failure probability $\delta \in (0,1)$, run BO with the Matern prior covariance function $k_{\mathrm{Matern}-\nu}(\boldsymbol{\theta}, \boldsymbol{\theta}')$ (Eq. 23) for $T = \mathrm{poly}(n)$ steps, where a scaling parameter $\eta_t$ for the acquisition function $\mathrm{UCB}_t(\boldsymbol{\theta})$ used in each step $t$ is predefined as $\eta_t = 2\log(\pi^2 t^2/(3\delta)) + 4p\log(4\pi pt^2 d^3 n^{7/2} q^{(d+1)p})$. If the parameter dimension $p$ is given by*

$$p \leq \mathcal{O}\left(\log n / \log(1/q)\right), \tag{86}$$

*the optimization error $\tilde{r}_T$ satisfies*

$$\tilde{r}_T \leq \mathcal{O}\left(\sqrt{p\log(pT^2 d^3 n^{7/2} q^{(d+1)p})}(\log T/T)^{\frac{\nu}{\nu+p}}\right) \tag{87}$$

*with a success probability of at least $1 - \delta$.*

*Proof.* Using the continuity property of the noisy PS-QNN objective function $\tilde{f}_q(\boldsymbol{\theta})$ as stated in Lemma 5.4 and a series of lemmas similar to Lemma C.2 to Lemma C.8, we can easily obtain

the aforementioned result. It is essential to emphasize the constraint imposed on the parameter dimension $p$. To guarantee the degree of discretization $\tau_t$ of at least 1, as mentioned in Lemma C.2, we need to discuss the range of $p$ that satisfies $pT^2d^3n^{7/2}q^{(d+1)p} \geq 1$. Since the number of steps $T = \text{poly}(n)$ and $p$ is at most $\text{poly}(n)$, we can establish the inequality

$$n^{c_2} \leq pT^2n^{7/2} \leq n^{c_1}, \tag{88}$$

where $c_1$ and $c_2$ are two very close constants. Then, we have

$$\frac{p}{n^{c_1}d^3} \leq \frac{1}{T^2n^{7/2}d^3} \leq \frac{p}{n^{c_2}d^3}. \tag{89}$$

Since $q \in (0,1)$, the relationship

$$q^{\frac{p(d+1)}{n^{c_2}d^3}} \leq q^{\frac{d+1}{T^2n^{7/2}d^3}} \tag{90}$$

holds. As $y^y$ is monotonically decreasing in the interval $(0, 1/e)$, we have

$$\left(\frac{1}{pT^2n^{7/2}d^3}\right)^{\frac{1}{pT^2n^{7/2}d^3}} \leq \left(\frac{1}{n^{c_1}d^3}\right)^{\frac{1}{n^{c_1}d^3}}. \tag{91}$$

Let

$$p \leq \frac{c_1 \log n + 3\log d}{(d+1)\log(1/q)n^{(c_1-c_2)}}, \tag{92}$$

then the above inequality implies

$$\left(\frac{1}{n^{c_1}d^3}\right)^{\frac{1}{n^{c_1}d^3}} \leq q^{\frac{p(d+1)}{n^{c_2}d^3}}, \tag{93}$$

which directly leads to

$$\left(\frac{1}{pT^2n^{7/2}d^3}\right)^{\frac{1}{pT^2n^{7/2}d^3}} \leq q^{\frac{d+1}{T^2n^{7/2}d^3}}, \tag{94}$$

that is $pT^2d^3n^{7/2}q^{(d+1)p} \geq 1$. Considering $d$ as a constant, Eq. 92 implies $p \leq \mathcal{O}(\log n/\log(1/q))$. $\square$

*Proof of Theorem 5.6.* Furthermore, when the strength $q \geq 1/\text{poly}(n)$, it is reasonable to treat $\log(pT^2d^3n^{7/2}q^{(d+1)p})$ as a constant. Therefore, our objective is to determine the effective $p$ that satisfies the condition $(p(\log(T)/T)^{\frac{\nu}{\nu+p}})^{1/2} \leq \epsilon$ with a constant threshold $\epsilon$. The previous result shows that $p \leq \tilde{\mathcal{O}}(\sqrt{\log n})$ and $T = \text{poly}(n^{1/\epsilon^2})$. Therefore, we have

$$p \leq \min\{\tilde{\mathcal{O}}(\sqrt{\log n}), \mathcal{O}(\log n/\log(1/q))\}. \tag{95}$$

Let $1/\text{poly}(n) \leq q \leq 1/n^{1/\sqrt{\log n}}$, then this constraint implies

$$\log n/\log(1/q) \leq \sqrt{\log n}, \tag{96}$$

that is $p \leq \mathcal{O}(\log n/\log(1/q))$. Thus, the proof of Theorem 5.6 is concluded. $\square$

# F NUMERICAL EXPERIMENTS

We perform numerical experiments in three directions: (1) employing a more diverse set of graph structures, (2) comprehensively comparing BO and GD, and (3) numerically validating our theoretical results. To ensure conceptual clarity, we first define three key concepts for solving the Maximum Cut problems using PS-QNN: (1) Exact Solution of the Problem as the exact Maximum Cut value of a given graph, (2) Circuit-Achievable Value as the maximum objective function value attainable with the PS-QNN at specified depth, and (3) Algorithm-Optimized Value as the optimized objective function value obtained via BO or GD.

Table 2: Performance of BO on diverse graph structures.

| Graph(Qubit=6) | 1 | 2 | 3 | 4 | 5 | 6 | 7 | 8 | 9 | 10 | Average |
|---|---|---|---|---|---|---|---|---|---|---|---|
| ExactSolution | 5 | 5 | 7 | 8 | 6 | 6 | 7 | 8 | 6 | 8 | 6.6 |
| AchievableValue | 4.27 | 4.58 | 5.91 | 6.83 | 5.2 | 5.49 | 6.53 | 6.98 | 5.57 | 7.54 | 5.89 |
| BO(Iteration=60) | 4.19 | 4.41 | 5.69 | 6.62 | 5.14 | 5.35 | 6.2 | 6.84 | 5.25 | 7.12 | 5.68 |

| Graph(Qubit=8) | 1 | 2 | 3 | 4 | 5 | 6 | 7 | 8 | 9 | 10 | Average |
|---|---|---|---|---|---|---|---|---|---|---|---|
| ExactSolution | 7 | 9 | 9 | 11 | 12 | 13 | 13 | 11 | 11 | 16 | 11.2 |
| AchievableValue | 6.27 | 7.84 | 8.13 | 9.83 | 10.11 | 11.05 | 11.76 | 10.11 | 10.45 | 13.69 | 9.93 |
| BO(Iteration=60) | 5.98 | 7.61 | 7.47 | 9.38 | 9.66 | 10.65 | 11.3 | 9.75 | 10.14 | 12.37 | 9.43 |

| Graph(Qubit=10) | 1 | 2 | 3 | 4 | 5 | 6 | 7 | 8 | 9 | 10 | Average |
|---|---|---|---|---|---|---|---|---|---|---|---|
| ExactSolution | 16 | 14 | 14 | 12 | 14 | 18 | 18 | 16 | 22 | 24 | 16.8 |
| AchievableValue | 13.43 | 11.07 | 12.19 | 10.65 | 12.48 | 14.91 | 15.31 | 14.41 | 20.13 | 20.78 | 14.54 |
| BO(Iteration=60) | 12.5 | 10.8 | 11.65 | 10.37 | 11.85 | 14.66 | 14.42 | 13.78 | 19.16 | 19.89 | 13.91 |

Table 3: Performance comparison of BO and GD.

| Graph(Qubit=10) | 1 | 2 | 3 | 4 | 5 | 6 | 7 | 8 | 9 | 10 | Average |
|---|---|---|---|---|---|---|---|---|---|---|---|
| ExactSolution | 16 | 14 | 14 | 12 | 14 | 18 | 18 | 16 | 22 | 24 | 16.8 |
| AchievableValue | 13.43 | 11.07 | 12.19 | 10.65 | 12.48 | 14.91 | 15.31 | 14.41 | 20.13 | 20.78 | 14.54 |
| BO(Iteration=30) | 12.40 | 10.61 | 11.26 | 10.33 | 11.78 | 14.47 | 14.31 | 13.59 | 18.44 | 19.71 | 13.69 |
| BO(Iteration=60) | 12.50 | 10.80 | 11.65 | 10.37 | 11.85 | 14.66 | 14.42 | 13.78 | 19.16 | 19.89 | 13.91 |
| BO(Iteration=90) | 12.67 | 10.80 | 11.78 | 10.37 | 11.86 | 14.66 | 14.42 | 13.78 | 19.35 | 19.96 | 13.97 |
| GD(Iteration=30) | 12.35 | 9.50 | 10.84 | 9.71 | 11.16 | 12.01 | 13.27 | 11.94 | 18.43 | 18.70 | 12.79 |
| GD(Iteration=60) | 12.37 | 9.48 | 10.86 | 9.90 | 11.23 | 12.56 | 13.28 | 12.25 | 18.42 | 18.94 | 12.93 |
| GD(Iteration=90) | 12.55 | 9.47 | 10.87 | 9.92 | 11.39 | 12.59 | 13.28 | 12.26 | 18.47 | 18.97 | 12.98 |

## F.1    PERFORMANCE OF BO ON DIVERSE GRAPH STRUCTURES

We investigate random graphs with 6, 8, and 10 vertices (10 graphs per size) and construct the Maximum Cut objective function using a 2-layer PS-QNN. For each graph, we run BO with 10 random initializations and 60 iterations per run. The results, summarized in Table 2, demonstrate that BO performs robustly, achieving average accuracies-defined as the ratio of the mean Algorithm-Optimized Value to the mean Exact Solution-of 86.06%, 84.20%, and 82.80% for graphs with 6, 8, and 10 vertices, respectively.

## F.2    PERFORMANCE COMPARISON OF BO AND GD

We comprehensively compare the performance of BO and GD by evaluating the Algorithm-Optimized Value and the number of steps to convergence. This comparison uses 10 randomly generated 10-vertex graphs, with the Maximum Cut objective function constructed for each graph using a 2-layer PS-QNN. To ensure a rigorous comparison, BO and GD are executed with 10 random initializations and tested for 30, 60, and 90 iterations. The results are summarized in Table 3.

Table 4: Numerical validation of theoretical results.

| Graph(Qubit=10) | 1 | 2 | 3 | 4 | 5 | 6 | 7 | 8 | 9 | 10 | Average |
|---|---|---|---|---|---|---|---|---|---|---|---|
| **ExactSolution** | 16 | 14 | 14 | 12 | 14 | 18 | 18 | 16 | 22 | 24 | 16.8 |
| **AchievableValue(depth=1)** | 12.46 | 9.73 | 11.13 | 9.91 | 11.73 | 13.54 | 14.02 | 13.45 | 18.90 | 20.21 | 13.51 |
| **Iteration($\epsilon$=0.9)** | 2 | 3 | 4 | 6 | 9 | 3 | 6 | 9 | 13 | 17 | 7.2 |
| **Iteration($\epsilon$=0.8)** | 3 | 4 | 4 | 7 | 9 | 4 | 6 | 9 | 19 | 18 | 8.3 |
| **Iteration($\epsilon$=0.7)** | 3 | 4 | 6 | 7 | 9 | 5 | 6 | 11 | 21 | 22 | 9.4 |
| **AchievableValue(depth=2)** | 13.43 | 11.07 | 12.19 | 10.65 | 12.48 | 14.91 | 15.31 | 14.41 | 20.13 | 20.78 | 14.54 |
| **Iteration($\epsilon$=2.0)** | 15 | 7 | 13 | 1 | 5 | 11 | 13 | 10 | 26 | 9 | 11 |
| **Iteration($\epsilon$=1.5)** | 19 | 12 | 17 | 8 | 14 | 19 | 18 | 20 | 36 | 18 | 18.1 |
| **Iteration($\epsilon$=1.0)** | 36 | 16 | 28 | 15 | 18 | 25 | 31 | 25 | 59 | 43 | 29.6 |

## F.3 NUMERICAL VALIDATION OF THEORETICAL RESULTS

Recognizing that error mitigation techniques can effectively address quantum circuit noise, we focus our analysis on the noiseless scenario. Our experiments use 10 randomly generated 10-vertex graphs, with the Maximum Cut objective function implemented via 1-layer and 2-layer PS-QNNs. For the 1-layer PS-QNN, we analyze the relationship between the optimization error $\epsilon$-defined as the difference between Circuit-Achievable Value and Algorithm-Optimized Value-and average iteration counts $T$ at error levels of 0.7, 0.8, 0.9. Similarly, for the 2-layer PS-QNN, we examine this relationship at error levels of 1, 1.5, 2. In both cases, we observe $\log T \propto 1/\epsilon^2$. The detailed results are summarized in Table 4.

## G THE USE OF LARGE LANGUAGE MODELS(LLMS)

During the preparation of this work, we use LLMs to assist in language polishing and editing the initial draft. This tool is used solely to improve grammatical fluency and sentence structure.

