# OpenReview forum: "A Theory of Training Parameter-Shared Quantum Neural Networks from a Bayesian Perspective"
_ICLR.cc/2026/Conference — ICLR 2026 Conference Withdrawn Submission_

### Official Review · Reviewer_kG5Y · 2025-10-22

**Soundness:** 1
**Presentation:** 2
**Contribution:** 2
**Rating:** 2
**Confidence:** 4

**Summary:**

The paper proposed a bayesian optimization perspective for optimizing PS-QNNs. The authors provide evidence for trainability of such models both in the noisy and noiseless regime.

There are several issues with nomenclature, and there are issues in the proofs (see below). Further, it has to be made clearer that these results hold under the assumption of local 1-designs. As it is communicated now, it seems to hold for all PS-QNNs and there is a missing discussion on whether and why this assumption holds for the architecture.

Further, it is unclear to me at this point why the authors choose to term their networks PS-QNN instead of sticking to the established QAOA framework (as a matter of fact, I also do not understand what is meant with "parameter-shared" as all parameters are only used once). They note in the introduction that QAOA is the commonly used term, and I did not find any other usage of PS-QNN upon a quick search.

The QAOA architecture is not changed to fit data or any ML setup, and, in fact, the benchmarks are also all based on optimization problems and actually have nothing to do with ML. I would therefore ask the authors to provide clarification on the nomenclature and why they think that the paper even fits the scope of this conference (noted here: https://iclr.cc/Conferences/2026).

**Strengths:**

The authors aim to provide a theoretical understanding of dynamics in variational optimization problems, which is a timely topic

**Weaknesses:**

- Please add references in Section 2. This seems to be taken from some prior work and it is not clearly stated where it is from (original QAOA paper?).
 - Naming of section 4 and 5.
 - Typo line 396: "vertice set".
 - Line 719: The proof that the expectation is zero is taken from the reference paper. The wording "we demonstrate" makes it seem like the authors' work. Please clarify.
- Please restate the Lemma in Section D again. Please also list the related Corollary in line 1068.
- The comparison of BO and GD in F.2 is lacking experimental details. Also, I strongly feel that this should be part of the main paper, since it is essentially a comparison to state-of-the-art. I also want to note that often the differences are only marginal, so I would expect some discussion on that and statistical significance tests. Moreover, I would expect to see the models trained multiple times, since the initial parameters can significantly impact the performance, giving evidence on the variance.
- F.3. Claim that error mitigation techniques effectively address quantum circuit noise. Circuit noise is one of the most significant issues in quantum computing and has to be addressed by error correction (otherwise state reference and justify). Also, a discussion of results is missing.

**Questions:**

- Why do you call the architecture PS-QNN if it is actually known as the quantum alternating operator ansatz? Further, why does the term not even occur in the reference that you state in line 37? Please clarify nomenclature and the difference to QAOA more clearly.
- Why are the results of Cerezo et al. (line 50) applicable to your setting? Could you elaborate on the differences in assumptions, since your architecture is fundamentally different, it seems odd to claim equivalence. Also, I would urge you to refrain from claiming they prove it for PS-QNNs, because they do not. (same goes through the subsequent lines, please be more precise about what is proven in the reference and how/why it applies to your specific architecture).
- What do you mean when you say (line 139), the statistical estimation of f can be achieved .... I assume you are talking about getting the empirical expectation, but the wording is imprecise.
- Why are there no references for Sections 3.1 and 3.2?
- Line 181: Why is this a strict equation? I assume you want y to be an unbiased estimator of f, but how can you strictly equate it with f plus a random noise variable. Should it not be just distributed accordingly?
- Equation 8: I'd suggest clarifying notation. It is a bit confusing to switch between $k_{t-1}(\theta)$ and $k_{t-1}(\theta, \theta')$. Do you come up with that yourself? Is this standard knowledge? I am missing a reference or derivation. Further, as far as I can see, $I_t$ is never introduced.
- What is the experimental setup for the results in Figure 1? How are $H_1$ and $H_2$ chosen? Figure b also needs further explanation on what is depicted and what the dots are.
- From where do you conclude that Gaussian processes are the widely preferred statistical model? I am missing references, and the experiments you conduct do not provide evidence for that statement either. Also, what are the alternatives?
- How can you employ the assumption 4.2? In the other works, they prove results based on this very condition, which hold if the architecture adheres to the assumption. You can prove your results under the assumption that it holds, but it requires some justification to actually claim that it holds true for PS-QNNs, which is essentially the conclusion of the work.
- Why is the adjoint of U missing in Fig2b (measurement arrow) and Eq.19? Also, there is a typo in Fig2a.
- Why is there a tilde over the O in Eq15? Can you quantify "with high success probability"?
- Elaborate on transition from Eq.31 to Eq.32. The proof is also inconclusive to me. Lemma B1 states that delta is not constant, but inherently dependent on the gradient variance (?), but in the proof of lemma 4.3 you claim you pick delta? You use the lemma in later proofs again (Lemma C.2) but seem to treat delta as a constant. There is a fundamental inconsistency here that needs to be addressed.
- It is unclear to me why it is called "Parameter-shared QNN". Maybe the authors could elaborate on what exactly is shared, since, each parameter occurs only once according to Eq 4?

---

### Official Review · Reviewer_VV1g · 2025-10-29

**Soundness:** 3
**Presentation:** 3
**Contribution:** 2
**Rating:** 4
**Confidence:** 3

**Summary:**

This paper develops a theoretical framework to study the convergence performance of training Parameter-Shared Quantum Neural Networks (PS-QNNs). Specifically, PS-QNNs are trained via the Bayesian optimization, which generates a new sample from an adaptively updated Gaussian process. The main results state that (1) noise-free PS-QNNs with depth scaling as roughly sqrt(log n) can be trained efficiently, and (2) noisy PS-QNNs, under local Pauli channels of strength q, are efficiently trainable up to depth about log n / log(1/q).

**Strengths:**

1. The paper introduces a gradient-free Bayesian-optimization framework for studying PS-QNN convergence.  The perspective is rarely explored in quantum machine learning, which bridges Bayesian inference and quantum variational optimization.

2. The key lemmas and theorems are stated clearly and logically consistent. The use of Gaussian-process smoothness and information-gain bounds is mathematically sound and connects well to the convergence analysis.

3. The paper directly addresses trainability and noise robustness, both crucial challenges for near-term quantum algorithms.

**Weaknesses:**

1. The theoretically trainable depth regime (sqrt(log n) or log n / log(1/q)) corresponds to very shallow circuits that can be efficiently simulated by classical tensor network methods. Hence, while the analysis is mathematically neat, its implications for achieving potential quantum advantage are questionable.

2. The numerical examples are small and only illustrative. In particular, the shallow PS-QNN circuits fail to find solutions close to the exact optima, suggesting that the convergence proven theoretically does not necessarily translate to useful performance.

3. Some refs are missing. For example, Ref.[1] provides the convergence analysis of training shallow QNNs with gradient descent.

[1] Girardi F, De Palma G. Trained quantum neural networks are gaussian processes[J]. Communications in Mathematical Physics, 2025, 406(4): 1-146.

**Questions:**

1. Can the framework be extended to deeper circuits, perhaps under weaker or more relaxed convergence guarantees?

2. How meaningful are the claimed polynomial scaling and convergence results, given that the global optima of shallow circuits are significantly different from the exact solutions?

---

### Official Review · Reviewer_4ENL · 2025-11-01

**Soundness:** 2
**Presentation:** 2
**Contribution:** 2
**Rating:** 4
**Confidence:** 3

**Summary:**

This paper studies the trainability issue of parameter-sharing quantum neural network (PS-QNN) from a theoretical perspective.The author proposes a Bayesian perspective, believing that the objective function of QNN can be regarded as sampled from a Gaussian Process, and thus uses Bayesian Optimization (BO) to analyze its convergence performance.

**Strengths:**

1.A Bayesian perspective is introduced to characterize the optimization landscape of QNN.
2.The article provides a lot of theoretical derivation to establish upper bounds on the optimization error, effective network depth, and the impact of noise, offering solid theoretical foundations for the proposed framework.
3.Providing a verifiable mathematical framework for "quantum trainability".

**Weaknesses:**

1.Lack of sufficient experimental verification. The full text is mainly theoretical derivation and hypothesis verification. There is insufficient experimental verification and it is difficult to evaluate the feasibility of the conclusions on actual quantum hardware. The tables in the appendix are not clearly expressed, and it is impossible to understand the author's meaning. Suggest a more detailed explanation of the table contents.
2.The numerical experiment part is not clearly stated clearly enough. The article only gives the experimental results, but does not explain the experimental settings in detail, including specific parameters, data sets, etc. In addition, there is a lack of clear standards for evaluating the quality of experimental results, and it is impossible to intuitively judge the effectiveness or advantages of the method. It is recommended to conduct quantitative evaluation or comparative analysis of the results to make the experimental conclusions more understandable and reproducible.

**Questions:**

1.How are experimental results judged? Are there clear convergence metrics or comparisons with other methods?
2.In the PS-QNN experiment with noise, how are the noise parameters selected?
3.How to reflect "trainability" in experiments? Are quantitative metrics provided for gradient, convergence speed, or optimization error?
4.Are the numerical results adequately visualized (convergence curves, gradient distributions, etc.) to intuitively support the theoretical conclusions?

---

### Official Review · Reviewer_xMiK · 2025-11-01

**Soundness:** 2
**Presentation:** 3
**Contribution:** 2
**Rating:** 2
**Confidence:** 4

**Summary:**

This paper develops a theoretical framework for analyzing the trainability of parameter-shared quantum neural networks (PS-QNNs) using Bayesian optimization (BO). The authors first study the noise-free case, establishing a Lipschitz-like continuity condition for the QNN objective and proving that efficient convergence can be achieved when the circuit’s effective parameter dimension scales as $O(\log(n))$. They then extend the analysis to the noisy setting, where local Pauli noise channels affect circuit performance. The study provides theoretical convergence guarantees and investigates the influence of noise on optimization landscapes. Finally, numerical experiments on Maximum Cut problems validate the theoretical scaling laws and confirm the efficiency of BO compared to gradient descent.

**Strengths:**

1.	The paper provides rigorous mathematical analysis of PS-QNN trainability, including proofs for convergence bounds and effective depth scaling.
2.	By combining Bayesian optimization with QNN analysis, it bridges classical machine learning techniques and quantum variational methods.
3.	Theoretical predictions are supported by numerical experiments on combinatorial optimization tasks such as Maximum Cut, demonstrating consistency between analytical results and simulation outcomes.

**Weaknesses:**

The paper suffers from an insufficient literature review. There is a substantial body of prior work on the convergence of quantum neural networks (QNNs), from both theoretical and experimental perspectives, that the authors have not taken into account [1–6]. Considering the existing findings regarding QNN convergence, I have several specific concerns about the authors’ results:

1.	Refs. [1-4] show that achieving global convergence generally requires overparameterization, meaning that the number of parameters in a QNN must exceed a certain threshold determined by the circuit architecture. In contrast, the authors claim that a circuit with depth $O(\log(n))$ suffices for rapid convergence to the global minimum with an approximation error $\epsilon$. This result appears inconsistent with the findings in Refs. [1-4], as $O(\log(n))$-depth circuits typically do not meet the overparameterization condition required for global convergence.
2.	A quantum circuit with depth $O(\log(n))$ has inherently limited expressivity. It is unclear how such a circuit could capture or represent the optimal solution landscape of the problem. Moreover, circuits of this depth can often be efficiently simulated on classical hardware, e.g., via Pauli-path [7] or tensor-network-based simulation methods [8]. In that case, exploring these circuits provides little to no quantum advantage, raising doubts about the practical relevance of the proposed approach.
3.	In prior studies, parameter sharing has been linked to the presence of symmetries in QNNs, which can significantly enhance their trainability. It would be valuable for the authors to clarify how their parameter-sharing setting differs from symmetry-based convergence enhancements reported in Refs. [5, 6]. Discussing this distinction could help position their contribution within the broader context of symmetry-driven QNN optimization.




[1] Larocca, Martin, et al. "Theory of overparametrization in quantum neural networks." Nature Computational Science 3.6 (2023): 542-551.

[2] You, Xuchen, Shouvanik Chakrabarti, and Xiaodi Wu. "A convergence theory for over-parameterized variational quantum eigensolvers." arXiv preprint arXiv:2205.12481 (2022).

[3] You, Xuchen, et al. "Analyzing convergence in quantum neural networks: deviations from neural tangent kernels." International Conference on Machine Learning. PMLR, 2023.

[4] Liu, Junyu, et al. "Analytic theory for the dynamics of wide quantum neural networks." Physical Review Letters 130.15 (2023): 150601.

[5] Wang, Xinbiao, et al. "Symmetric pruning in quantum neural networks." arXiv preprint arXiv:2208.14057 (2022).

[6] Sauvage, Frederic, et al. "Building spatial symmetries into parameterized quantum circuits for faster training." Quantum Science and Technology 9.1 (2024): 015029.

[7] Rudolph, Manuel S., et al. "Classical surrogate simulation of quantum systems with LOWESA." arXiv preprint arXiv:2308.09109 (2023).

[8] Pan, Feng, and Pan Zhang. "Simulation of quantum circuits using the big-batch tensor network method." Physical Review Letters 128.3 (2022): 030501.

**Questions:**

The questions are included in the Weakness.

---

### Note · Authors · 2025-11-13

I have read and agree with the venue's withdrawal policy on behalf of myself and my co-authors.